theoretical biology/biomathematics/mathematical modelling

upscaling, micro–macro evolution, eco-evolutionary dynamics, moment closure

**Authors for correspondence:**
Jan Martin Nordbotten
e-mail: jan.nordbotten@math.uib.no
Nils Chr. Stenseth
e-mail: n.c.stenseth@ibv.uio.no

# The dynamics of trait variance in multi-species communities

Jan Martin Nordbotten[1], Folmer Bokma[2],
Jo Skeie Hermansen[2] and Nils Chr. Stenseth[2]

[1]Department of Mathematics, University of Bergen, 5020 Bergen, Norway
[2]Centre for Ecological and Evolutionary Synthesis (CEES), Department of Biosciences, University of Oslo, 0316 Oslo, Norway

 JMN, 0000-0003-1455-5704; FB, 0000-0002-0049-2127;
JSH, 0000-0003-1589-4273; NCS, 0000-0002-1591-5399

In this paper, we establish the explicit connection between deterministic trait-based population-level models (in the form of partial differential equations) and species-level models (in the form of ordinary differential equations), in the context of eco-evolutionary systems. In particular, by starting from a population-level model of density distributions in trait space, we derive what amounts to an extension of the typical models at the species level known from adaptive dynamics literature, to account not only for abundance and mean trait values, but also explicitly for trait variances. Thus, we arrive at an explicitly polymorphic model at the species level. The derivations make precise the relationship between the parameters in the two classes of models and allow us to distinguish between notions of fitness on the population and species levels. Through a formal stability analysis, we see that exponential growth of an eigenvalue in the trait covariance matrix corresponds to a breakdown of the underlying assumptions of the species-level model. In biological terms, this may be interpreted as a speciation event: that is, we obtain an explicit notion of the blow-up of the variance of (possibly a linear combination of) traits as a precursor to speciation. Moreover, since evolutionary volatility of the mean trait value is proportional to trait variance, this provides a notion that species at the cusp of speciation are also the most adaptive. We illustrate these concepts and considerations using a numerical simulation.

# 1. Introduction

A range of models loosely termed 'adaptive dynamics' have been introduced to study the interaction between ecology and evolution, and hence link population dynamics to evolutionary

**Figure 1.** Categorization of models for coupled ecological and evolutionary interaction. Modified after [1]. Blue arrows indicate previously established formal relationships between model categories; the red arrow indicates the establishment of a direct formal relationship between deterministic population-level and deterministic species-level models, which is the aim of the current paper.

dynamics. In a review of such adaptive dynamics models, Dieckmann *et al*. [1] categorized them along two axes: deterministic versus stochastic, and monomorphic versus polymorphic (their fig. 8.2). Since we are explicitly interested in polymorphic species-level models (a key component of the present contribution), however, it becomes natural to instead categorize models along the two axes of deterministic versus stochastic, and population versus species (figure 1). In this figure, and throughout the paper, we abbreviate ordinary and partial differential equations as ODEs and PDEs, respectively.

Formal relationships between some of these model categories have been established, in particular between the stochastic and the deterministic models: when population-level stochastic models are ensemble-averaged to a deterministic equation, this is generally achieved using the framework of the Fokker–Planck equations (for an introduction, e.g. [2], and for a review in a biological context, see [3]). When species-level stochastic equations are ensemble-averaged [4], this typically follows the framework of stochastic differential equations (e.g. [5] and a recent review in a biological context [6]). However, to our knowledge, and in accordance with Dieckmann *et al*.'s review of the literature [1], no direct link has been established between deterministic models at the population and species levels, including both population dynamics and evolutionary change. While many studies consider evolutionary dynamics in the context of equilibrium ecosystems, it has recently been pointed out that fully coupled eco-evolutionary models are essential even when addressing long-term evolutionary trends [7].

To establish the connection between a population-level model and a species-level model, it is natural to consider the population-level model as 'ground truth', and to employ a formal upscaling method to derive the species-level model. (For a review of upscaling methodologies, see [8].) Here, we use a population-level model previously presented by two of the authors [9] as a starting point, as this model is conceptually similar to other models of similar kind discussed in the literature (e.g. [10]), yet more general. Through a precise definition of the species concept, and by deriving moment equations for each species, we directly arrive at species-level equations, the structure of which resembles that of established monomorphic models at the species level [4,11]. However, our upscaling results in a model which is polymorphic, with explicit equations governing the evolution of trait variances (cf. [4]). Moreover, we obtain explicit expressions for all parameters in the species-level model in terms of parameters at the population level.

Establishing such a formal relationship between models at these different scales is needed to justify species-level models, because the natural unit of most ecological and evolutionary processes is the individual or population and not the species. Moreover, it follows from the establishment of an explicit relationship between two different models for the same system that these models should be *consistent*, i.e. they should lead to the same system behaviour in the cases where both models are valid. Therefore, we analyse the species-level model, in view of the assumptions used in its derivation, to explicitly state the conditions under which it breaks down. This breakdown can be given two interpretations: either the

upscaling is simply insufficiently general to cover this case or the model becomes fundamentally inapplicable. We argue that these interpretations are both correct: the species-level model breaks down when the assumption of clearly defined species is invalid, which is a fundamental assumption of the upscaling as well as the premise under which a species-level model is relevant. (When this assumption is not met, the prudent modeller will revert to a population-level model to analyse the system dynamics.)

We note that an objective similar to that of the current paper was pursued by Débarre *et al.* [12]. In their paper, they derive from a population-level model, the species-level equations for the change in the mean and variance of populations in the absence of population dynamics, which is sufficient when discussing conditions for evolutionary stationary equilibria. We instead allow for varying population sizes, which leads to qualitatively and quantitatively different equations on the species level. Our resulting species-level equations form a closed system of ODEs that allow for the joint consideration of ecological and evolutionary interactions. We also point out the related work by Sasaki & Dieckmann [13], who consider the evolution of separate morphs within the same species. Their approach, however, does not directly generalize to the case of distinct species.

The remainder of this work is structured as follows: §2 provides the main theoretical result of the paper, namely the derivation of the deterministic species-level models from the deterministic population-level model. Section 3 provides an analysis of the stability of the species-level model, and the interpretation of the breakdown of this model. In §4, we consider a numerical example, validating the upscaling, as well as illustrating possible dynamics of trait variance during speciation. Finally, in §5, we discuss the main results and their implications.

# 2. Upscaling a population-level model to the species level

We subdivide this section into two parts, in order to first present the population-level model. To make the upscaling accessible to a broader audience, we keep the exposition formal (from a mathematical perspective), and relatively simple in the sense of avoiding technicalities associated with functional analysis. Thus, we invoke the blanket assumption that all functions are sufficiently smooth for the expressions to be well defined.

## 2.1. Population-level model

As a starting point, we adopt a trait-specific population-level model that uses a quite general equation for modelling the ecological and evolutionary dynamics of asexual haploid individuals, in the absence of spatial variability [9]. It is based on the following non-local and nonlinear PDE describing the population growth rate, governing the dynamics of the abundance density, $n = n(x, t)$

$$\frac{\partial n}{\partial t} = rn - bn^2 + n \int_\Omega \alpha(x,x')n(x')\mathrm{d}x' + \nabla \cdot (g \nabla n), \tag{2.1}$$

where $t$ is time and the coordinates $x$ indicate a position in $d$-dimensional phenotype space $\Omega$ (i.e. $x \in \Omega \subset \mathbb{R}^d$). Thus, each coordinate $x$ corresponds to a specific set of $d$ phenotypic trait values and the distance between any two coordinates measures the difference between the two sets of phenotypic trait values in the domain $\Omega$. Thus, the abundance density $n(x, t)$ can be explicitly defined as the number of individuals per measure on $\mathbb{R}^d$, at some time $t$. The parameter $r = r(x)$ is the inherent *per capita* growth rate (i.e. a net measure of births and deaths for an individual with trait values $x$ at infinitesimal abundance). The term $bn^2 = b(x)n(x, t)^2$ is a local self-interaction term (i.e. it limits an individual based on other members of the population that have identical trait values). We will refer to this self-limitation term as 'individual self-limitation', to set it apart from species self-limitation, which is an emergent property defined in §2.2. The parameter $\alpha(x, x')$ is the interaction coefficient representing the effect individual members of the population with trait values $x'$ have on individuals with trait values $x$. It will emerge in §2.2 that this coefficient governs not only interaction between species, but also interactions within species. Finally, $g = g(x)$ is an intergenerational trait diffusion coefficient, which allows the trait values of the next generation to differ from those of the current, and thus represents in a mean sense the stochasticity of the underlying process at the individual level. The model defined by equation (2.1) is the starting point for the current study, but other population-level models, in particular stochastic models, could also be analysed using the same formalism as detailed below (e.g. [12]). The right-hand side of equation (2.1) can after division by $n$ informally be considered as the fitness function at the population level.

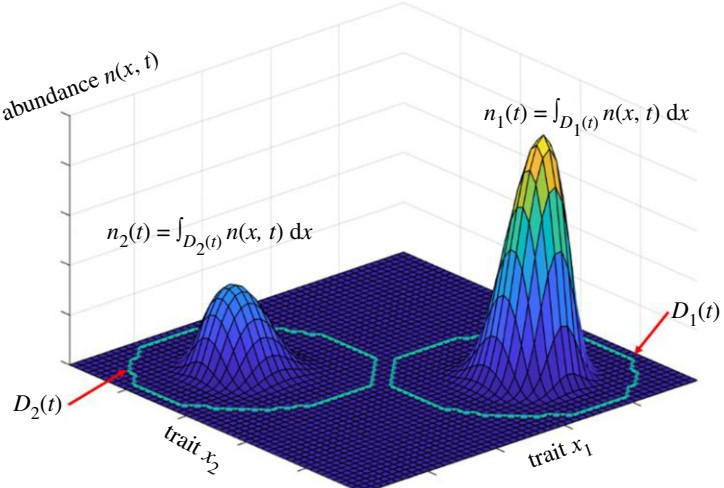

**Figure 2.** An abundance density $n(x,t)$, showing two species in the case of $d=2$ traits. The outer boundaries of the species domains $D_1(t)$ and $D_2(t)$ are indicated by curves.

Regarding the model parameters, equation (2.1) is well posed with quite few restrictions on the parameters [9]. In particular, we note that the individual self-limitation $b(x)$ does not need to be positive (only non-negative), thus, depending on the structure of the interaction coefficient $\alpha$, one can consider equation (2.1) without the individual self-limitation term.

## 2.2. Upscaling to a species-level model

We consider equation (2.1) to be the governing equation for the ecological interaction and evolution of a distribution of a population in phenotype space, and will conduct a formal upscaling procedure in the spirit of moment expansions.

To derive a species-level model, we will make the natural *a priori* assumption that the solution distribution $n(x,t)$ of equation (2.1) is such that $S$ species can be uniquely defined as subpopulations with non-overlapping trait values. More specifically, we require that there exist exactly $S$ closed domains $D_i(t) \subset \Omega$, for $i=1\dots S$, such that at any point in time the domains $D_i$ are non-overlapping, $D_i(t) \cap D_j(t) = \emptyset$, and such that $n(x,t)=0$ if $x$ is not in any $D_i(t)$. While in nature distributions in the values of one or a few traits often overlap between species, we believe that this is a realistic assumption if sufficiently large numbers of traits are considered. Moreover, we discuss the failure of this assumption in the case of speciation events in §§3 and 4. Now, the subpopulation within domain $D_i$ is considered as the composition of species $i$, so that we can uniquely define the abundance $n_i(t)$ of the $i$th species as the integral of the abundance density over $D_i(t)$ (figure 2)

$$n_i(t) = \int_{D_i(t)} n(x,t)\,\mathrm{d}x. \tag{2.2}$$

We will need the vector of the mean trait values $x_i(t)$, and the matrix $\sigma_i^2(t)$ of trait variances (and covariances), for each species $i$. Both of these will be time-dependent, due not only to the trait values, and hence region $D_i(t)$ possibly evolving through time, but also due to intraspecific changes in abundance density $n(x,t)$. These trait means and variances are naturally defined as the first and second moments of $n(x,t)$ within $D_i(t)$

$$x_i(t) = \frac{1}{n_i(t)} \int_{D_i(t)} n(x,t)x\,\mathrm{d}x \tag{2.3}$$

and

$$\sigma_i^2(t) = \frac{1}{n_i(t)} \int_{D_i(t)} n(x,t)(x - x_i(t))^2\,\mathrm{d}x. \tag{2.4}$$

Note that we interpret the product of vectors as the outer product $\otimes$, unless the inner (dot) product is explicitly stated. That is, for two vectors $a$ and $b$ with components $a_{l_1}$ and $b_{l_2}$, $ab = (a \otimes b)_{l_1 l_2} = a_{l_1} b_{l_2}$ and similarly $a^2 = aa$.

From equations (2.1) and (2.2), we can then obtain the ecological species dynamics by taking zero-order moments over $D_i$, which yields (for details, see appendix A.4.1)

$$\frac{\mathrm{d}n_i}{\mathrm{d}t} = R_i(x_i, \sigma_i, t)n_i - B_i(x_i, \sigma_i, t)n_i^2 + n_i \sum_{j \neq i} A_{i,j}(x_i, x_j, \sigma_i, \sigma_j, t)n_j. \tag{2.5}$$

This equation is structurally similar to a Lotka–Volterra model [14], but deviates in one important respect: the terms of the equation contain not only the mean trait values $x_j$ of each species, but also the variances $\sigma_i^2$. Indeed, for the growth rate, we obtain

$$R_i(x_i, \sigma_i, t) \equiv r_i + \frac{1}{2}\nabla\nabla r_i : \sigma_i^2, \tag{2.6}$$

where subscripts indicate that a function is evaluated at the mean trait value of that species (for details, see appendix A.1)

$$r_i(t) \equiv r(x_i(t), t) \quad \text{and} \quad \nabla\nabla r_i \equiv (\nabla\nabla r)(x_i(t), t), \text{ etc.} \tag{2.7}$$

Here, and later, we have used the double dot to indicate the double contraction of two tensors, generalizing the dot product of vectors. Precisely, for two tensors $a$ and $b$ with components $a_{i,j,\ldots k_1, l_1}$ and $b_{k_2, l_2, \ldots m, n}$, then $(a:b)_{i,j,\ldots m,n} = \sum_{l,k} a_{i,j,\ldots k,l} b_{k,l,\ldots m,n}$. In the particular case where $a$ and $b$ are two tensors (matrices), this expression simplifies to $(a:b) = \text{tr } ab$, where $ab$ is the normal matrix product and the trace of a matrix is defined as the sum of its elements on the main diagonal.

We digress for a moment with a qualitative observation that is valid when the interaction between species is weak: consider the case of a system with a single species, near its equilibrium. In equation (2.5), the '$B$'-term captures self-limitation and is negative or zero. The '$R$'-term describes 'self-growth minus death', and must be positive for a single species to exist. Anticipating that for a single species at equilibrium, the net production term associated with $r_i$ is in general near a maximum, the curvature $\nabla\nabla r_i$ will in this instance have negative eigenvalues, and thus the last term of equation (2.6) is negative. The variance appearing in the expression for $R_i$ then implies a trade-off between species optimization (low variance, higher growth rate) and species resilience (lower growth rate, but higher variance and hence greater potential for adaptive change).

Just like the species growth rate, also the species self-limitation term depends on the variances of traits. As derived in appendix A.4.1, it appears as a balance between individual self-limitation and intraspecific competition (or cooperation if that term is positive) as

$$B_i(x_i, \sigma_i, t) \equiv \frac{1}{\sqrt{(2\pi)^d}|\sqrt{2}\sigma_i|}\left(b_i + \frac{1}{4}\nabla\nabla b_i : \sigma_i^2\right) - (\alpha_{i,i} + \nabla_i\nabla_i\alpha_{i,i} : \sigma_i^2). \tag{2.8}$$

The variance plays a similar role as above in impacting the self-limitation term: if a species is near a minimum of the self-limitation term, then the Hessian matrix $\nabla\nabla b_i$ will have positive eigenvalues, and there is again a trade-off, as higher variance leads to more self-limitation, and vice versa.

Finally, the interspecific interactions are primarily governed by the mean trait values, as manifested by $\alpha_{i,j}$, but also by the trait variances of the species itself, and (naturally) the trait means and variances of the species it interacts with

$$A_{i,j}(x_i, x_j, \sigma_i, \sigma_j, t) \equiv \alpha_{i,j} + \frac{1}{2}\nabla_i\nabla_i\alpha_{i,j} : \sigma_i^2 + \frac{1}{2}\nabla_j\nabla_j\alpha_{i,j} : \sigma_j^2. \tag{2.9}$$

A common assumption in the literature is that $\alpha(x, x')$ is negative and symmetric with respect to the arguments. We note that this does not, however, lead to a simplification of equation (2.9), since $\nabla_i\nabla_i\alpha_{i,j}$ does not inherit the symmetry.

We emphasize that in the derivations used to obtain the preceding equations, we have consistently used the assumption from the beginning of the section that the parameter functions are sufficiently smooth, and therefore, a Taylor expansion is applicable locally within each $D_i$. By contrast, the normal distribution is used only as an approximate closure model, not as an assumption. In particular, it is only invoked to obtain the constants in front of the terms containing $b_i$ in equation (2.8), and the derivations of equations (2.6), (2.7) and (2.9) do not involve any use of the normal distribution. If, say, a uniform distribution on a $d$-dimensional hyper-sphere was used as closure assumption, we would

obtain a slightly perturbed $B_i^*$ with coefficients (see appendix A.2.2)

$$B_i^*(x_i,\sigma_i,t) \equiv \frac{1}{\mathcal{M}_d(v_d)|\sigma_i|}\left(b_i + \frac{1}{2}\nabla\nabla b_i : \sigma_i^2\right) - (\alpha_{i,i} + \nabla_i\nabla_i\alpha_{i,i} : \sigma_i^2),$$

where $\mathcal{M}_d(v)$ is the volume of a $d$-dimensional ball of radius $v$ and $v_d$ is the radius of the sphere that has unit covariance. As an example, we thus see that for a two-dimensional trait space, the leading coefficient in the definition of $B_i$ varies depending on the closure assumption by a factor

$$\frac{\sqrt{(2\pi)^2}\sqrt{2}}{\pi(\sqrt{8})^2} = \frac{\sqrt{2}}{4} \approx 0.35.$$

The parameters $r_i$, $b_i$ and $\alpha_{i,j}$ are species-specific, since they depend on $x_i$ (and $x_j$). This implies that the ecological system (2.5) is directly coupled to the evolution of the species themselves. Therefore, we need equations that describe the evolution of species traits. These can be derived from the first moment of equation (2.1) by a Taylor series expansion of the coefficients $r$, $b$ and $\alpha$ locally around the coordinates $x_i$, followed by application of the same procedure as for equation (2.5) as detailed in appendix A.4.2. This yields

$$\frac{dx_i}{dt} = \sigma_i^2 \cdot \left(\nabla r_i - \frac{n_i}{2\sqrt{(2\pi)^d}|\sqrt{2}\sigma_i|}\nabla b_i + \sum_j \nabla_i\alpha_{i,j} \cdot n_j\right). \tag{2.10}$$

Equation (2.10) is similar to equation (4.12) in Dieckmann & Law [4], except for a factor ½ in their equation. This is a consequence of their definition of variance, which is not the variance of the traits in the species, but rather the variance of the process at the stochastic level (similar to our parameter $g$). Thus, equation (2.10) is reminiscent of the univariate Lande equation [15,16], which states that evolutionary change in a trait over a single generation is directly proportional to the product of the additive genetic variance $V_a$ (additive genetic variance–covariance matrix in the multivariate form) and the selection gradient (vector of selection gradients in the multivariate form). However, the term multiplying $\sigma_i^2$ corresponds to the lowest-order terms of the species fitness gradient (defined as the right-hand side of equation (2.5) divided by $n_i$) only if the individual self-limitation term $b$ has negligible variation. If the individual self-limitation term cannot be neglected, the determinant of the standard deviation $|\sigma_i|$ appears in the denominator in equation (2.10). This implies that trait evolution is not strictly proportional to variance, but may saturate somewhat at high variances. Indeed, this separates our derivation from that of Débarre *et al.* [12] (see their equation (4)), who obtain exactly the product of covariance and a selection gradient. Our observation that species trait evolution is not in general directly proportional to a fitness gradient leads us to de-emphasize this terminology.

It is important to note that equations (2.5) and (2.10) do not form a complete model without an equation for the change of trait variances $\sigma_i^2$. To obtain equations for the temporal evolution of trait variances, we used derivations and assumptions similar to those used above for the evolution of trait means, namely taking second-order moments of equation (2.1) within $D_i$ (appendix A.4.3)

$$\frac{d\sigma_i^2}{dt} = V_{0,i} + V_{1,i}\sigma_i^2 + \sigma_i^2 V_{2,i}\sigma_i^2. \tag{2.11}$$

In equation (2.11), we have identified three main drivers of trait variances. First, there is the commonly understood impact of intergenerational increase in trait variance due to imperfect inheritance, given by

$$V_{0,i} = 2g_i. \tag{2.12}$$

Second, individual self-limitation will tend to increase trait variance, since self-limitation discourages identical traits within a species. This is reflected in the term

$$V_{1,i} = \frac{n_i}{2\sqrt{(2\pi)^d}|\sqrt{2}\sigma_i|}\left(b_i + \frac{1}{2}(\sigma_i^2 : \nabla\nabla b_i)\right). \tag{2.13}$$

In contrast with the two first terms, the third term typically decreases the trait variance, as it is related in some sense to the curvature of the population fitness and requires a moment closure approximation (e.g. [17,18], and also a discussion in [13])

$$V_{2,i} = \nabla\nabla r_i + \sum_j \nabla_i\nabla_i\alpha_{i,j}n_j - \frac{n_i\nabla\nabla b_i}{4\sqrt{(2\pi)^d}|\sqrt{2}\sigma_i|}. \tag{2.14}$$

The balance between these three drivers of trait variance is critical for understanding the long-term stability of a species. Most notably, if the final component $V_{2,i}$ has positive eigenvalues, as may occur near saddle-points in the population fitness function, then the trait variance may grow exponentially and result in speciation.

While there are significant differences in the details, equations similar to equation (2.11) have been proposed as far back as Lande [19], and have also appeared in later works (e.g. Débarre *et al.* [12]). Our results differ from previous results in several respects. Most importantly, our results are applicable in a dynamical context. This impacts directly the concrete expressions derived above. A full literature survey is beyond the scope of this paper, but we will point out the main differences between our work and Débarre *et al.*, as it is recent, and their aims and scope are most closely related to ours. In particular, we note that in equation (2.11), the dynamics of trait variances depend on three terms, and that equation (2.14) is not simply the Hessian matrix of a fitness measure (i.e. $V_{2,i}$ is neither the Hessian of the right-hand side of (2.1) nor (2.4)). Both of the above are in contrast with the results of Débarre *et al.*, who do not include $V_0$ or $V_1$, and whose definition of $V_2$ is the Hessian of the population fitness function.

While our results apply to a dynamical context, our model does not explicitly consider demographic stochasticity. It has been shown (e.g. [20]) that under certain conditions, demographic stochasticity may qualitatively and quantitatively affect the behaviour of the kind of processes modelled here. It is generally thought that such conditions are small population size and/or pronounced stochasticity: especially under such circumstances, the dynamics of real-world systems may differ from the predictions of the model presented here.

Jointly, equations (2.5), (2.10) and (2.11) form a closed system of ODEs, for the abundance, mean trait values and trait variances of the species in a system. To our knowledge, this represents the first unified derivation of these equations in their complete form.

# 3. Stability of species models

Equations (2.5), (2.10) and (2.11) are derived directly from the population-level eco-evolutionary equation (2.1), invoking three assumptions: firstly, that species can be uniquely defined, and secondly that the parameters depend continuously on traits so that the gradients in (2.10) and (2.14) are well defined, and so that the characteristic parameters can be well approximated by the parameter value of the mean trait. Thirdly, we invoke a moment closure relationship to obtain precise factors for all terms involving the parameter $b$, and in obtaining (2.14). We truncate the model at trait variances, in the sense that we assume that skewness and excess kurtosis in trait value distributions can be neglected.

Near stationary evolutionary equilibria, the curvature terms $V_{2,i}$ will have negative eigenvalues, thus balancing the natural increase of trait variance otherwise driven by $V_{0,i}$. However, negative eigenvalues of the curvature of $V_{2,i}$ should not be taken for granted: away from evolutionary equilibria, the curvature of growth factors $\nabla\nabla r_i$ may have some positive eigenvalues, potentially leading to an exponential blow-up of equation (2.11). Clearly, if the variance for a species $i$ becomes unbounded then a bounded domain $D_i(t)$ cannot be defined, which invalidates our definition of the species itself. While unbounded growth of an eigenvalue of the covariance invalidates the derivation of the species-level model, it does not impact the well definedness of the population-level model given in equation (2.1) [9]. Therefore, we interpret the blow-up of variance as a case of model failure (as opposed to invalid parameters) and postulate that a *blow-up of trait variance in the species-based model is an indicator of a speciation event in the population-based model*. This is conceptualized in figure 3. The observation that the eigenvalues of the Hessian matrix of fitness impact evolutionary stability and speciation has been exploited in previous work (e.g. [1,13,21]). The explicit expressions for the dynamics of variance embodied in equations (2.11–2.14) allow for this statement to be made precise in mathematical terms.

# 4. Numerical examples

We consider two numerical examples: first, a validation of the correspondence between the population-level and species-level models, and then an exploration of the breakdown of the species-level model. These two numerical examples thus corroborate the analytical results of §§2 and 3, respectively.

In this section, since we will be making numerous references to the two models, we will use the abbreviation PLM for the population-level model, defined by equation (2.1), and we will use the abbreviation SLM for the species-level model, defined by equations (2.5), (2.10) and (2.11).

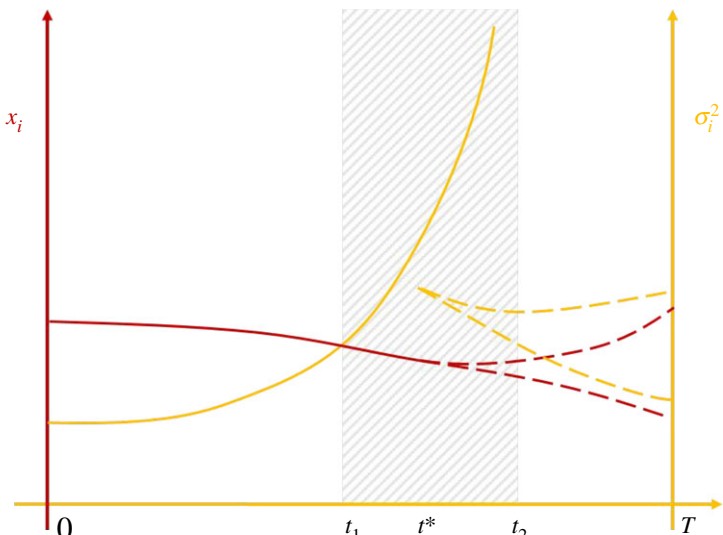

**Figure 3.** A conceptual illustration of a speciation event. At early times, a species can clearly be defined with the mean trait $x_i$ (red, left axis) and variance $\sigma_i^2$ (yellow, right axis). The shaded part of the figure represents the period of speciation, and also breakdown of the species model. Here, the trait variance grows exponentially, and the identification of a single species eventually breaks down. At later times, once sufficient divergence exists in trait space to identify two domains $D_i$, a two species-model can be justified (dashed lines), with diverging traits and moderate trait variances.

The numerical examples are calculated based on Matlab code, which is available as electronic supplementary material.

## 4.1 Validation of the upscaling from population-level to species-level models

We provide a simple example of a species adapting to an evolutionary optimum as a validation of the SLM derived above, as well as the numerical methods used to implement it.

For this case, consider the idealized setting of two traits, each scaled to the unit square, thus $d = 2$ and $\Omega = [0, 1]^2$. Moreover, for $x \in \Omega$, we consider the parameters

$$r(x,t) = 1 - \left\| x - \left[\frac{1}{2}, \frac{1}{2}\right] \right\|^2, \quad b(x,t) = 10^{-3}, \quad \alpha(x,x') = -1, \quad g(x) = 2 \cdot 10^{-5}.$$

Here, we use the Euclidian vector norm. We initialize the population as a normal distribution of unit mass, centred at $x_0(t = 0) = [1/4, 1/4]$, with isotropic variance $\sigma_1^2(t = 0) = 10^{-3}I$. Note that this initial condition is not in equilibrium.

For this system, we solve both equation (2.1) and equations (2.5), (2.10) and (2.11). The PLM equation (2.1) is resolved by finite differences on a $100 \times 100$ Cartesian grid, with an exponential integrator for the linear terms and operator splitting for the differential terms. We used the built-in ODE integrator ode15s in Matlab to resolve the ordinary differential equations comprising the SLM.

The solution is shown for $t = 1000$ in figure 4. Here, three plots are shown. The bottom plot is the reference solution $n(x, t)$ of the PLM, and represents in this context the true state of the system, in the sense of the solution to equation (2.1). The middle level of the plot is based on the statistics of the true state: we plotted a normal distribution with the obtained statistics of $n(x, T)$. That is, if we let $\mu_0(t) = \int_\Omega n(x,t)\,dx$, $\mu_1(t) = 1/\mu_0(t) \int_\Omega n(x,t)\,x\,dx$ and $\mu_2(t) = 1/\mu_0(t) \int_\Omega n(x,t)\,(x - \mu_1(t))^2\,dx$ then the second level plots $N(x; \mu_0(t), \mu_1(t), \mu_0(t))$. This then represents the best possible solution that can be represented by only an abundance, mean trait vector and variance matrix. Note that the shape of the true distribution differs markedly from that of a normal distribution. Finally, the top level of the figure shows the result of the species-level model represented by equations (2.5), (2.10) and (2.11), i.e. a normal distribution given by $N(x; n_i(t), x_i(t), \sigma_i(t))$. Comparison of the top and middle levels provides an assessment of the accuracy of the SLM. Comparison of the middle and bottom level is an assessment of the adequacy of representing a species in terms of only abundance, mean trait vector and variance matrix.

A more quantitative time-dependent comparison is shown in figure 5, where only the species statistics are shown, as functions of time. Note how the PLM and SLM correspond during the initial

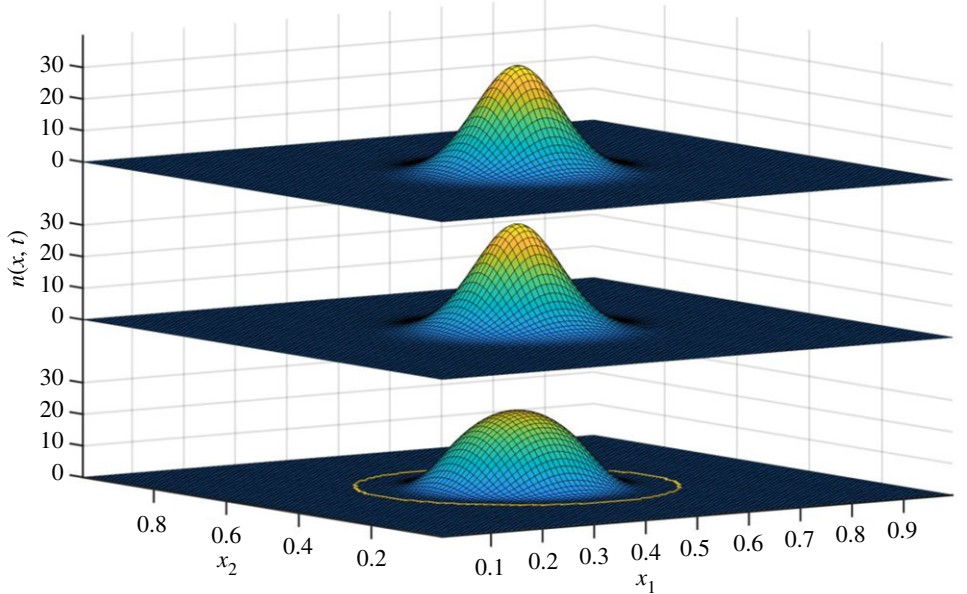

**Figure 4.** Population distribution in trait space for $t = 1000$. From bottom to top: the PLM, the species-level interpretation of the PLM and the SLM. The yellow circle is the region within which $n(x, t) > 0$, i.e. the domain of $D_1(t)$. The vertical axis measures abundance, while the horizontal axes represent traits. For a full description, see the main text.

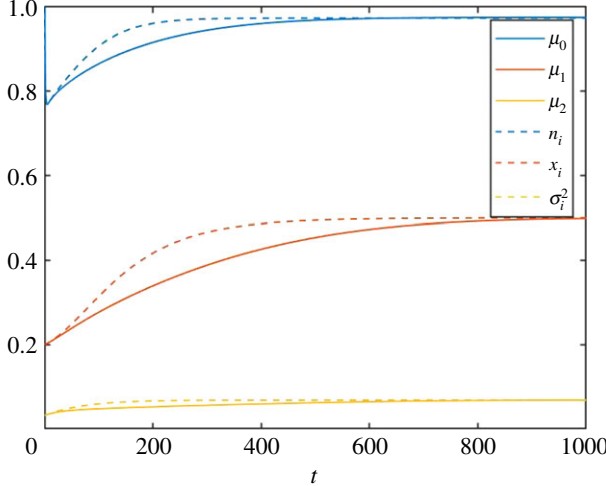

**Figure 5.** Time evolution of abundance (blue), first component of trait vector (red) and the first component of the variance matrix (yellow) for the PLM (solid lines) and the SLM (dashed lines).

ecological equilibration. At intermediate times, the models show fair, but not excellent agreement. At still later time, the models coincide again. The mismatch at intermediate times is due to the moment closure approximation (equation 2.14): since the skewness of the intraspecific distribution of trait values cannot be captured by the mean and variance of traits, its influence on the evolution of the mean trait values is neglected, which is of modest importance during transitions as evidenced by the non-normal distribution seen in figure 4 (see also figure 9 below).

## 4.2. Exploration of trait variance before and after a speciation event

In order to highlight the importance of accurately modelling changes in trait variances, we consider a case where a single species undergoes a speciation event due to changes in its external environment. To keep the discussion simple, we assume one of the simplest possible non-trivial set-ups that allows

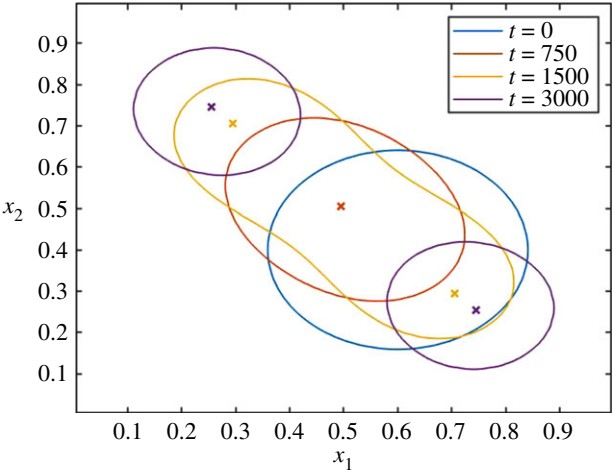

**Figure 6.** Illustration of a speciation event as calculated by the PLM, and induced by a bifurcation in the maximum for the trait-dependent growth rate $r(x,t)$ at $t = 1000$ as explained in the text. The figures show the solution of equation (2.1), plotted as the curves where abundance $n(x,t) = 10^{-4}$. Initially (blue and red curve), only a single species can be defined, while eventually (purple curve), two distinct species can be defined. At the time of the yellow curve, the transition is in progress. The maximum of $r(x,t)$ is marked by an '$x$' for each $t$, but the maximum for $t = 0$ (blue) is invisible because it coincides with that for $t = 750$ (red).

for speciation, namely, a two-trait system similar to that used in §4.1. The precise biological interpretation of these traits is not important, we simply suppose that they have been non-dimensionalized and scaled so that they vary between 0 and 1, and that correspondingly the trait domain is constrained as $\Omega = [0,1] \times [0,1] \subset \mathbb{R}^2$. We constrain the dynamics of the system to appear as a consequence of the trait-dependent growth rate $r(x,t)$: we keep all other parameters constant (see appendix B for detailed description of the parameters) but choose a growth rate that leads to a population fitness whose shape gradually changes over time. This serves as a proxy for either external forcing (such as climate) or the existence of a third (dominant) species that changes the conditions for the species we model. Initially ($t=0$), the growth rate $r(x,t)$ has a single peak, which over time becomes gradually broader, forming a ridge, eventually leading to a saddle point at $t = 1000$ where the centre of the ridge at $[1/2, 1/2]$ is no longer the optimum, and two optima start to appear, at either end of the ridge. These optima gradually diverge and the valley between them becomes deeper and broader until at $t = 2000$ the optima reach the points $[3/4, 1/4]$ and $[1/4, 3/4]$ and no further external dynamics are imposed on the system.

The expected behaviour is that a single species will exist for $t < 1000$. The initial condition is away from the eventual saddle point, and during this initial phase, the species will thus move in trait space. During this period, we also observe an elongation of the abundance in trait space along the ridge, i.e. along the eigenvector that changes sign at the saddle point at $t = 1000$. After this time, a speciation event may occur. To explore how this speciation event is captured by the PLM and SLM, we initialize the system with a single species near the middle of the domain. By solving equation (2.1), we can obtain the reference dynamics for this system, which is illustrated for selected times in figure 6.

The same scenario can now be analysed using the SLM, utilizing an *a priori* assumption of a single species. In figure 7, the solution is plotted in terms of the abundance, the position in trait space and the first component of the standard deviation $\sigma_i$. Figure 7 shows that as the speciation event approaches, the single-species model breaks down, and indeed becomes invalid when $t = 1171$. Thus, the variance blow-up is indeed a precursor, and a marker, of the approaching speciation event and the failure of the single-species assumption. Indeed, the variance starts increasing significantly already well before $t = 1000$, exaggerating the effect seen in the PLM in figure 6. This is qualitatively in accordance with equation (2.11).

The natural extension is to consider a two-species model for the same system. We initialize the SLM model as above, but now consider two species. Both species are initialized with the same mean and variance, consistent with the single-species case (see appendix B for details). The resulting dynamics are shown in figure 8, and figure 9 shows the situation at $t = 2000$. Figure 8 illustrates how the trait variance peaks around the speciation event also under the *a priori* assumption of two independent species.

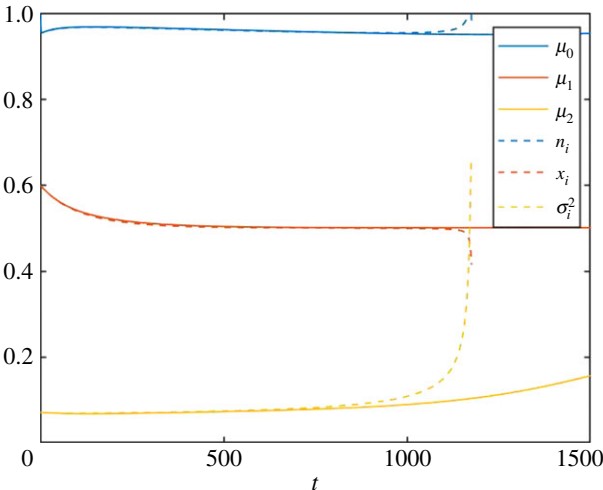

**Figure 7.** Abundance (blue), mean trait position (red) and first component of the trait variance matrix (yellow), as calculated from the solution of the PLM (solid lines), and the SLM (dashed lines) with the *a priori* assumption of a single species. As detailed in the text, the trait-dependent growth rate $r(x, t)$ has a single maximum for $t < 1000$, and two gradually moving maxima for $1000 < t \leq 2000$, after which they remain constant.

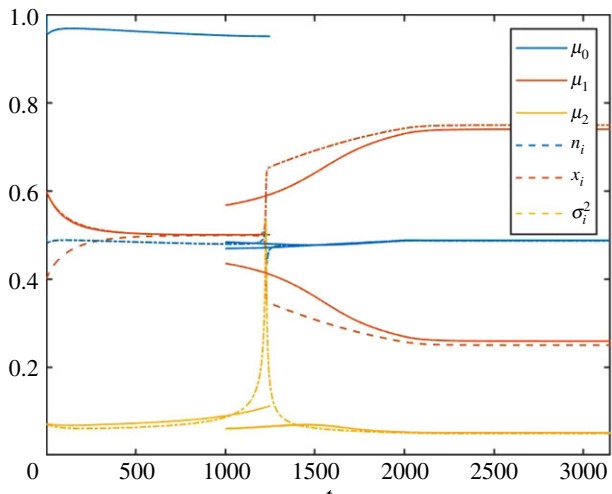

**Figure 8.** Temporal evolution of total species abundance (blue), mean trait values (red) and first component of trait variance matrix (yellow) for the PLM (solid lines) and the SLM (dashed and dash-dotted lines), across a speciation event: the trait-dependent growth rate $r(x, t)$ has a single maximum for $t < 1000$, and two gradually moving maxima for $1000 < t \leq 2000$, after which they remain constant, as detailed in the text. This is the same scenario as in figure 7, but here it is assumed *a priori* that two species exist, instead of a single species as in figure 7. However, the mean and variance of traits are reported for a single-species interpretation of the PLM for $t \leq 1500$, and for a two-species interpretation for $t \geq 1000$. Note that before the speciation event, the two species essentially converge to what must be interpreted as a single species, and thereafter sharply diverge in the mean trait value (even though due to the set-up of the problem, their abundance and trait variance functions overlap).

This rise in trait variance is then not only a precursor to speciation, but also implies that the trait divergence of the two species is accelerated immediately after the speciation event, when compared with what would be expected based on the general trait variance observed at earlier and later times.

Figure 9 illustrates that although it is a fair approximation to consider species in terms of their abundance, trait means and variances, the implied normal distributions somewhat deviate from the results of the PLM. However, the slight differences in the predictions of evolutionary rates of mean trait values introduced by this discrepancy will probably be obscured by the uncertainties introduced when considering any real system.

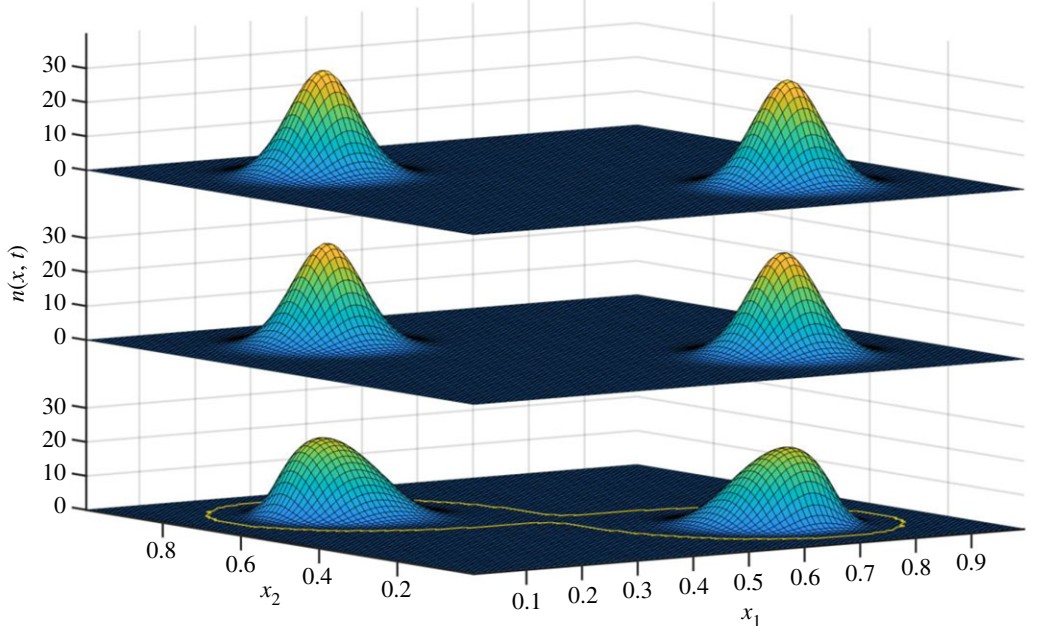

**Figure 9.** Comparison of two models as in figure 4, but for a case 2, at a time ($T = 2000$) where two species can be recognized, as described in the text. The bottom level is the PLM, and the middle layer is the species-level interpretation of PLM in terms of two species. The top level is the SLM with two species. The yellow line delimits the region within which $n(x, T) > 0$, i.e. the domain of $D_1(T)$: we see that even at this state of divergence, the assumption of two independent species is not (yet) fully justified.

This example emphasizes the importance of incorporating speciation events in SLMs. However, this implies that the system of ODEs must be augmented in a way that allows for additional variables to be introduced during the modelling time-window. Since this augmentation is exactly at what is essentially a bifurcation point in the original system (e.g. [22]), it is to be expected that the resulting model predictions may be highly sensitive to the exact way the original system is augmented. A full understanding of speciation events and their incorporation into SLMs is a topic of current research.

## 5. Conclusion

We have established a direct link between deterministic models at two levels of eco-evolutionary modelling: at the finer level, a population model based on a (non-local) partial-differential equation for the abundance density in trait space, at which level no explicit concept of a species exists. At the coarser level, a species model represented by a family of ordinary differential equations governing the abundance, mean trait values and trait variances of explicitly defined species. In contrast with previous work, our derivation of the species-level model from the population-level model leads directly to a population dynamical and polymorphic description at the species level, thus incorporating both ecology and evolution. There is a close relationship between the model derived here and other models in the literature: if trait variances are assumed not to evolve but to remain constant, and if there is no individual self-limitation ($b = 0$), then equations (2.5) and (2.10) become similar to the monomorphic model derived by Dieckmann & Law [4]. If also the trait means are assumed to remain constant, then equation (2.5) reduces to the multi-species Lotka–Volterra model for abundance in the absence of evolution [14]. Furthermore, our equation for species variances is also a generalization of the results obtained by Débarre *et al.* [12]. Our numerical examples illustrate and validate our theoretical results, and illustrate how explicitly modelling trait variance of each species is essential for correctly capturing the evolution implied by the system.

We see from equation (2.14) that a critical process occurs if the matrix $V_{2,i}$ does not have negative eigenvalues. This loosely corresponds to the notion of a saddle point in population fitness, in particular if there is no individual self-limitation ($b = 0$). At that moment, the trait variance may blow up, which can lead to either of two outcomes: the first possible scenario is that the species, by virtue of its increased variance, rapidly evolves along the selection gradient as described by equation (2.10).

The second possibility is speciation, as illustrated in figure 3. Which of these two possible outcomes occurs will depend on the exact path the species follows in trait space towards the saddle point.

By contrast, a species that resides at a local maximum in fitness will tend to have a relatively smaller variance, as given by a balance of the terms in equation (2.11). Because the rate of evolutionary response in a trait of a species is to first-order proportional to trait variance $\sigma_i^2$ (equation 2.10), we see that such a species will not be able to adapt as rapidly, should external factors alter the system. That is also the case if the population itself changes over time (due to climate change or similar processes): species near (moving) local optima, and having smaller trait variances, will have a slower response to changes in environmental conditions than species with variances that were inflated by the curvature of growth factors $\nabla\nabla r_i$ away from these optima.

Data accessibility. The numerical examples shown in this manuscript can be reproduced by Matlab code available as electronic supplementary material.

Authors' contributions. J.M.N. and N.C.S. conceived the study. All authors took part in shaping the analysis and results presented in the work. J.M.N. drafted the manuscript. All authors critically revised the content and approved the final version.

Competing interests. We declare we have no competing interests.

Funding. The research was funded in part by Norwegian Research Council project no. 263149.

Acknowledgements. The authors wish to thank Mats Brun, Vikash Pandey, Jostein Starrfelt and Kjetil Lysne Voje for constructive discussions on the topic of this manuscript.

# Appendix A: Derivation of species equations

The species equations are obtained by moment closure within each domain $D_i(t)$, together with second-order Taylor expansions in phenotype space of the functions $r(x, t)$, $b(x, t)$ and $\alpha(x, x', t)$.

We first summarize the following important tools.

## A.1. Taylor expansions

The Taylor series expansion of a multidimensional function around a point $x_i$ is given as

$$r(x,t) = r(x_i,t) + \nabla r(x_i,t) \cdot (x - x_i) + \frac{1}{2}\nabla\nabla r(x_i,t) : (x - x_i) \otimes (x - x_i) + \mathcal{O}(|x - x_i|^3).$$

We will consistently assume that terms proportional to $|x - x_i|^3$ are negligible, and, therefore, only keep the three first terms of the series. Moreover, we use the shorthand notation

$$r_i \equiv r(x_i,t), \quad \nabla r_i \equiv \nabla r(x_i,t) \quad \text{and} \quad \nabla\nabla r_i \equiv \nabla\nabla r(x_i,t).$$

Moreover, we will use the shorthand for the outer vector product

$$(x - x_i)^n \equiv (x - x_i) \otimes \cdots \otimes (x - x_i).$$

A similar Taylor series expansion is then used for $b(x, t)$, thus we use the approximation that, for $x \in D_i(t)$,

$$b(x,t) \approx b_i + \nabla b_i \cdot (x - x_i) + \frac{1}{2}\nabla\nabla b_i : (x - x_i)^2.$$

On the other hand, for $\alpha(x, x', t)$, we need to consider the expansion in both phenotypical arguments, i.e. for points $x \in D_i(t)$ and $x' \in D_j(t)$, we have

$$\alpha(x,x',t) = \alpha(x_i,x_j,t) + \nabla_x\alpha(x_i,x_j,t) \cdot (x - x_i) + \frac{1}{2}\nabla_x\nabla_x\alpha(x_i,x_j,t) : (x - x_i)^2 + \dots$$

$$\cdots + \nabla_{x'}\alpha(x_i,x_j,t) \cdot (x' - x_j) + \frac{1}{2}\nabla_{x'}\nabla_{x'}\alpha(x_i,x_j,t) : (x' - x_j)^2 + \dots$$

$$\cdots + \nabla_x\nabla_{x'}\alpha(x_i,x_j,t) : (x - x_i) \otimes (x' - x_j) + \mathcal{O}(|x - x_i|^3).$$

Again, we use a more compact notation, and omit terms proportional to $|x - x_i|^3$ or higher, to obtain

$$\alpha(x,x',t) \approx \alpha_{i,j} + \nabla_i \alpha_{i,j} \cdot (x - x_i) + \frac{1}{2} \nabla_i \nabla_i \alpha_{i,j} : (x - x_i)^2 + \cdots$$

$$\cdots + \nabla_j \alpha_{i,j} \cdot (x' - x_j) + \frac{1}{2} \nabla_j \nabla_j \alpha_{i,j} : (x' - x_j)^2 + \nabla_i \nabla_j \alpha_{i,j} : (x - x_i) \otimes (x' - x_j).$$

We will for convenience consider the genetic mutation term to vary sufficiently little that it is appropriate to consider a single-term approximation, i.e. we will consistently use the approximation

$$g(x, t) \approx g_i(t) \equiv g(x_i(t), t).$$

This simplification has no impact on the resulting discussion.

# A.2. Moments of squared distributions

We will frequently need to approximate the term in equation (1.1). This requires a closure assumption, i.e. an assumption that within, then is by assumption well approximated by a known distribution. The most natural choice will be the normal distribution; however, a contrasting choice is the uniform distribution on an interval. Thus, we need to know the moments of these distributions squared.

## A.2.1. Moments of the squared normal distribution

Let $k$ be a positive integer. Then it holds that for a normal distribution that

$$\int_{\mathbb{R}^d} N^2(x; n_i, x_i, \sigma_i) (x - x_i)^k \, dx = \frac{n_i^2}{(2\pi)^d |\sigma_i|^2} \int_{\mathbb{R}^d} \exp\left( -\frac{x^T \cdot \sigma_i^{-2} \cdot x}{2} \right)^2 x^k \, dx$$

$$= \frac{n_i^2}{\sqrt{(2\pi)^d} |\sqrt{2}\sigma_i|} \frac{1}{\sqrt{(2\pi)^d} \left|\frac{\sigma_i}{\sqrt{2}}\right|} \int_{\mathbb{R}^d} \exp\left( -\frac{x^T \cdot (\sigma_i/\sqrt{2})^{-2} \cdot x}{2} \right) x^k \, dx$$

$$= \frac{n_i^2}{\sqrt{(2\pi)^d} |\sqrt{2}\sigma_i|} \begin{cases} 0 & \text{if} \quad k \text{ is odd} \\ 1 & \text{if} \quad k = 0 \\ \frac{1}{2}\sigma_i^2 & \text{if} \quad k = 2 \\ 2^{-k/2}\mu_{k,i}(N) & \text{if} \quad k \geq 4 \end{cases}.$$

Here, $\mu_{k,i}(N)$ is the $k$th standardized moment (of the normal distribution $N(x; n_i, x_i, \sigma_i)$).

## A.2.2. Moments of the uniform distribution squared

In order to appreciate the impact of the choice of closure model, we will also consider the uniform distribution within a hyper-sphere. Recalling that the Heaviside function $H(\xi)$ evaluates to zero for negative values and 1 for positive values of $\xi$, we define the multivariate uniform distribution with given moments $U(x; n_i, x_i, \sigma_i)$ as

$$U(x; n_i, x_i, \sigma_i) = \frac{n_i}{\mathcal{M}_d(v_d) |\sigma_i|} H\left( 1 - \frac{(x - x_i)^T \cdot \sigma_i^{-2} \cdot (x - x_i)}{v_d^2} \right).$$

Here, $v_d$ is the radius of a $d$-dimensional ball such that the mean variance is the identity tensor $I_d$, i.e. $v_d$ is the number such that

$$\int_{B(v_d)} x^2 \, dx = I_d \int_{B(v_d)} 1 \, dx = I_d \mathcal{M}_d(v_d),$$

where $B(v_d)$ is a $d$-dimensional ball of radius $v_d$, and $\mathcal{M}_d(v_d)$ its volume. As examples, we can calculate that for a single dimension $v_1 = \sqrt{3}$ and for two dimensions $v_2 = \sqrt{8}$.

A direct calculation now gives that the $k$th moment of the uniform distribution squared satisfies

$$\int_{\mathbb{R}^d} U^2(x;n_i,x_i,\sigma_i)\,(x-x_i)^k\,\mathrm{d}x = \frac{n_i^2}{\mathcal{M}_d^2(v_d)|\sigma_i|^2}\int_{\mathbb{R}^d} H\left(1-\frac{x^{\mathrm{T}}\cdot\sigma_i^{-2}\cdot x}{v_d^2}\right)x^k\,\mathrm{d}x$$

$$= \frac{n_i^2}{|\sigma_i|}\begin{cases} 0 & \text{if} \quad k \text{ is odd} \\ 1 & \text{if} \quad k = 0 \\ \sigma_i^2 & \text{if} \quad k = 2 \\ \mu_{k,i}(U) & \text{if} \quad k \geq 4 \end{cases},$$

where $\mu_{k,i}(U)$ is the $k$th standardized moment of the uniform distribution $U(x; n_i, x_i, \sigma_i)$.

## A.3. Kurtosis and Isserli's theorem

Due to the product of a three-term Taylor expansion and the calculation of variance, we will encounter the fourth-order moment, also known as kurtosis. In particular, we will use the standardized fourth moment, which is a four-tensor $\mu_{4,i}$, where the four components $l_1 \ldots l_4$ of the tensor $\mu_{4,i}$ can be approximated based on the variance using Isserli's theorem

$$(\mu_{4,i})_{l_1,l_2,l_3,l_4} \approx (\sigma_i^2)_{l_1,l_2}(\sigma_i^2)_{l_3,l_4} + (\sigma_i^2)_{l_1,l_3}(\sigma_i^2)_{l_2,l_4} + (\sigma_i^2)_{l_1,l_4}(\sigma_i^2)_{l_2,l_3}.$$

This expression is exact for normal distributions. It is clear that the approximation of the fourth moment, being based on second moments in the manner above, contains significant symmetries. Moreover, if we let $J$ be a symmetric second-order tensor, this allows us to obtain the simplified expression

$$\mu_{4,i} : J \approx (\sigma_i^2 : J)\sigma_i^2 + 2\sigma_i^2 J \sigma_i^2.$$

An alternative to Isserli's theorem is to use moment closure based on the uniform distribution (see §A.2.2); however, we will not elaborate this here.

## A.4. Derivation of species equations

Using the preparation of §§A.1–A.3, the derivation of species equations is a straightforward application of the definition of moments and equation (2.1).

### A.4.1. Equations for species abundance

We are interested in the temporal change of species abundance, i.e.

$$\frac{\mathrm{d}n_i}{\mathrm{d}t} = \frac{\mathrm{d}}{\mathrm{d}t}\int_{D_i(t)} n(x,t)\,\mathrm{d}x = \int_{D_i(t)} \frac{\partial n}{\partial t}(x,t)\,\mathrm{d}x$$

$$= \int_{D_i(t)} rn - bn^2 + n\int_\Omega \alpha(x,x')n(x')\,\mathrm{d}x' + \nabla\cdot(g\nabla n)\,\mathrm{d}x.$$

Here, we have used that $n(x,t) = 0$ on $\partial D_i$ by the assumption that species can be defined, and substituted equation (1.1) for the rate of change of $n(x,t)$. We proceed by approximating each term.

*Species growth rate*

$$\int_{D_i(t)} rn\,\mathrm{d}x \approx \int_{D_i(t)} \left(r_i + \nabla r_i\cdot(x-x_i) + \frac{1}{2}\nabla\nabla r_i : (x-x_i)^2\right)n\,\mathrm{d}x$$

$$= \left(r_i + \frac{1}{2}\nabla\nabla r_i : \sigma_i^2\right)n_i(t).$$

*Individual self-limitation*

$$\int_{D_i(t)} bn^2\,\mathrm{d}x \approx \int_{D_i(t)} \left(b_i + \nabla b_i\cdot(x-x_i) + \frac{1}{2}\nabla\nabla b_i : (x-x_i)^2\right)n^2\,\mathrm{d}x$$

$$\approx \frac{1}{\sqrt{(2\pi)^d}|\sqrt{2}\sigma_i|}\left(b_i + \frac{1}{4}\nabla\nabla b_i : \sigma_i^2\right)n_i^2(t).$$

*Ecological interactions*

$$\int_{D_i(t)} n \int_\Omega \alpha(x,x')n(x')\,\mathrm{d}x'\,\mathrm{d}x = \int_{D_i(t)} n \sum_j \int_{D_j(t)} \alpha(x,x')n(x')\,\mathrm{d}x'\,\mathrm{d}x$$

$$\approx \int_{D_i(t)} n \sum_j \int_{D_j(t)} \left( \begin{array}{c} \alpha_{i,j} + \nabla_i \alpha_{i,j} \cdot (x - x_i) + \frac{1}{2}\nabla_i\nabla_i\alpha_{i,j} : (x - x_i)^2 + \cdots \\ \cdots + \nabla_j\alpha_{i,j}\cdot(x'-x_j) + \frac{1}{2}\nabla_j\nabla_j\alpha_{i,j}:(x'-x_j)^2 + \nabla_i\nabla_j\alpha_{i,j}:(x-x_i)\otimes(x'-x_j) \end{array} \right) n(x')\,\mathrm{d}x'\mathrm{d}x$$

$$= \int_{D_i(t)} n \sum_j \int_{D_j(t)} \left( \alpha_{i,j} + \frac{1}{2}\nabla_i\nabla_i\alpha_{i,j}:(x-x_i)^2 + \frac{1}{2}\nabla_j\nabla_j\alpha_{i,j}:(x'-x_j)^2 \right) n(x')\,\mathrm{d}x'\mathrm{d}x$$

$$= n_i \sum_j \left( \alpha_{i,j} + \frac{1}{2}\nabla_i\nabla_i\alpha_{i,j}:\sigma_i^2 + \frac{1}{2}\nabla_j\nabla_j\alpha_{i,j}:\sigma_j^2 \right) n_j.$$

*Genetic mutations*

$$\int_{B_i(t)} \nabla \cdot (g\nabla n)\,\mathrm{d}x = 0.$$

Combining the above calculations, we obtain the following expanded Lotka–Volterra-type system

$$\frac{\mathrm{d}n_i}{\mathrm{d}t} = R_i(x_i,\sigma_i,t)n_i - B_i(x_i,\sigma_i,t)n_i^2 + n_i \sum_{j\neq i} A_{i,j}(x_i,x_j,\sigma_i,\sigma_j,t)n_j,$$

where the species growth rate depends both on the traits and their variance, and is defined as

$$R_i(x_i,\sigma_i,t) \equiv r_i + \frac{1}{2}\nabla\nabla r_i : \sigma_i^2.$$

The species self-limitation depends on both individual self-limitation and intraspecific competitive interactions, and also depends on both traits and their variance as

$$B_i(x_i,\sigma_i,t) \equiv \frac{1}{\sqrt{(2\pi)^d}\left|\sqrt{2}\sigma_i\right|}\left( b_i + \frac{1}{4}\nabla\nabla b_i : \sigma_i^2 \right) - (\alpha_{i,i} + \nabla_i\nabla_i\alpha_{i,i} : \sigma_i^2).$$

Finally, inter-species interactions depends not only the traits of both species, but the variance of both species

$$A_{i,j}(x_i,x_j,\sigma_i,\sigma_j,t) \equiv \alpha_{i,j} + \frac{1}{2}\nabla_i\nabla_i\alpha_{i,j} : \sigma_i^2 + \frac{1}{2}\nabla_j\nabla_j\alpha_{i,j} : \sigma_j^2.$$

## A.4.2. Equations for species traits

In the same manner as above, we calculate for the first moment of each species

$$n_i\frac{\mathrm{d}x_i}{\mathrm{d}t} = \frac{\mathrm{d}n_i x_i}{\mathrm{d}t} - x_i\frac{\mathrm{d}n_i}{\mathrm{d}t} = \frac{\mathrm{d}}{\mathrm{d}t}\int_{D_i(t)} n(x,t)x\,\mathrm{d}x - x_i\frac{\mathrm{d}}{\mathrm{d}t}\int_{D_i(t)} n(x,t)\,\mathrm{d}x = \int_{D_i(t)} \frac{\partial n}{\partial t}(x,t)\,(x-x_i)\,\mathrm{d}x$$

$$= \int_{D_i(t)} \left( rn - bn^2 + n\int_\Omega \alpha(x,x')n(x')\mathrm{d}x' + \nabla\cdot(g\nabla n) \right)(x-x_i)\,\mathrm{d}x.$$

Again, we resolve each term as

$$\int_{D_i(t)} rn(x-x_i)\,\mathrm{d}x \approx \int_{D_i(t)} \left( r_i + \nabla r_i\cdot(x-x_i) + \frac{1}{2}\nabla\nabla r_i:(x-x_i)^2 \right) n(x-x_i)\,\mathrm{d}x$$

$$= n_i\sigma_i^2\cdot\nabla r_i + \frac{1}{2}\,\mu_{3,i}:\nabla\nabla r_i.$$

In accordance with our representation of a species in terms of the mean, position and variance, we will use the normal distribution to approximate higher-order moments, by Isserli's theorem, and consequently the odd moments vanish. In the following, we will, therefore, omit the skew term $\mu_{3,i}$.

By a similar calculation, we then obtain

$$\int_{D_i(t)} bn^2 (x - x_i) \, dx \approx \int_{D_i(t)} n^2 (x - x_i)^2 \cdot \nabla b_i \, dx \approx \frac{n_i^2 \sigma_i^2}{2\sqrt{(2\pi)^d} |\sqrt{2}\sigma_i|} \cdot \nabla b_i$$

and

$$\int_{D_i(t)} n \sum_j \int_{D_j} \alpha(x,x')n(x') \, dx' \, (x - x_i) \, dx$$

$$\approx \int_{D_i(t)} n \sum_j \int_{D_j(t)} \nabla_i \alpha_{i,j} \cdot (x - x_i) \, n(x') \, dx'(x - x_i) \, dx$$

$$= n_i \sigma_i^2 \cdot \sum_j \nabla_i \alpha_{i,j} n_j.$$

It remains true that the Laplacian term evaluates to zero in the first moment, and thus we summarize the dynamics of the mean trait as

$$\frac{dx_i}{dt} = \sigma_i^2 \cdot \left( \nabla r_i - \frac{n_i}{2\sqrt{(4\pi)^d}|\sigma_i|} \nabla b_i + \nabla_i \sum_j \alpha_{i,j} \, n_j \right).$$

Note that in contrast with common expectation, this expression shows that when there is some individual self-limitation in the system, i.e. when $b > 0$, then the change in trait need not be strictly proportional to the trait variance, since the trait variance also appears in the denominator of the second term in the parenthesis.

## A.4.3. Equations for species trait variance evolution

Our derivation of the expanded Lotka–Volterra and adaptive dynamics systems have shown the important, and non-trivial, role that trait variance plays in both ecological and evolutionary dynamics. Indeed, we will see that the dynamics of trait variance itself is a critical factor in species evolution. As such, we do not consider trait variance a species property, but rather derive explicit equations for its temporal change.

By the same procedure as above, we then obtain

$$\frac{d}{dt}(n_i \sigma_i^2) = \frac{d}{dt} \int_{D_i(t)} n(x - x_i)^2 \, dx = \int_{D_i(t)} \frac{\partial n}{\partial t} (x - x_i)^2 \, dx - 2\frac{dx_i}{dt} \int_{D_i(t)} n(x - x_i) \, dx$$

$$= \int_{D_i(t)} \left( rn - bn^2 + n \int_\Omega \alpha(x,x')n(x') \, dx' + \nabla \cdot (g\nabla n) \right) (x - x_i)^2 \, dx - 2\frac{dx_i}{dt} n_i(x_i - x_i).$$

Again, each term in the integral can be approximated by a similar procedure as previously. From the growth term, we obtain

$$\int_{D_i(t)} rn(x - x_i)^2 \, dx \approx \int_{D_i(t)} \left( r_i + \nabla r_i \cdot (x - x_i) + \frac{1}{2}\nabla\nabla r_i : (x - x_i)^2 \right) n(x - x_i)^2 \, dx$$

$$= n_i r_i \sigma_i^2 + n_i \mu_{3,i} \cdot \nabla r_i + \frac{n_i}{2} \mu_{4,i} : \nabla\nabla r_i \approx n_i r_i \sigma_i^2 + \frac{n_i}{2}((\sigma_i^2 : \nabla\nabla r_i)\sigma_i^2 + 2\sigma_i^2 \nabla\nabla r_i \sigma_i^2).$$

Here, we have used the Gaussian distribution to approximate the fourth moments on a matrix, as a way of moment closure [18], i.e.

$$\mu_{4,i} : \nabla\nabla r_i \approx (\sigma_i^2 : \nabla\nabla r_i)\sigma_i^2 + 2\sigma_i^2 \nabla\nabla r_i \sigma_i^2.$$

However, other choices of moment closure are also possible, for a discussion in the biological context, see [13]. Similarly, from the individual self-limitation term

$$\int_{D_i(t)} bn^2 (x - x_i)^2 \, dx \approx \int_{D_i(t)} \left( b_i + \frac{1}{2} \nabla\nabla b_i : (x - x_i)^2 \right) n^2 (x - x_i)^2 \, dx$$

$$= \frac{n_i^2}{\sqrt{(2\pi)^d} \left| \sqrt{2}\sigma_i \right|} \left( \frac{\sigma_i^2 b_i}{2} + \frac{\mu_{4,i} : \nabla\nabla b_i}{8} \right)$$

$$\approx \frac{n_i^2}{2\sqrt{(4\pi)^d} |\sigma_i|} \left( \sigma_i^2 b_i + \frac{1}{4}(\sigma_i^2 : \nabla\nabla b_i)\sigma_i^2 + \frac{1}{2} \sigma_i^2 \nabla\nabla b_i \sigma_i^2 \right).$$

The competitive terms leads to

$$\int_{D_i(t)} \left( n \int_\Omega \alpha(x,x')n(x') \, dx' \right) (x - x_i)^2 \, dx$$

$$\approx \int_{D_i(t)} n \sum_j \int_{D_j} \left( \alpha_{i,j} + \frac{1}{2} \nabla_i \nabla_i \alpha_{i,j} : (x - x_i)^2 + \frac{1}{2} \nabla_j \nabla_j \alpha_{i,j} : (x' - x_j)^2 \right) n(x') \, dx' \, (x - x_i)^2 \, dx$$

$$= \sum_j n_j \int_{D_i(t)} \left( \alpha_{i,j} + \frac{1}{2} \nabla_i \nabla_i \alpha_{i,j} : (x - x_i)^2 + \frac{1}{2} \nabla_j \nabla_j \alpha_{i,j} : \sigma_j^2 \right) n(x)(x - x_i)^2 \, dx$$

$$= n_i \sigma_i^2 \sum_j \left( \alpha_{i,j} + \frac{1}{2} \nabla_j \nabla_j \alpha_{i,j} : \sigma_j^2 \right) n_j + \mu_{4,i} : \sum_j \frac{1}{2} \nabla_i \nabla_i \alpha_{i,j} n_j$$

$$\approx n_i \sigma_i^2 \sum_j \left( \alpha_{i,j} + \frac{1}{2} \nabla_j \nabla_j \alpha_{i,j} : \sigma_j^2 \right) n_j + \frac{n_i}{2} \left( \left( \sigma_i^2 : \left( \sum_j \nabla_i \nabla_i \alpha_{i,j} n_j \right) \right) \sigma_i^2 + 2\sigma_i^2 \left( \sum_j \nabla_i \nabla_i \alpha_{i,j} n_j \right) \sigma_i^2 \right).$$

Finally, we integrate by parts twice on the differential term to obtain (here $I$ is the identity matrix)

$$\int_{D_i(t)} \nabla \cdot (g\nabla n) (x - x_i)^2 \, dx = 2n_i g_i I.$$

Combining these calculations, we obtain

$$\frac{d}{dt} n_i \sigma_i^2 = n_i r_i \sigma_i^2 + \frac{n_i}{2}((\sigma_i^2 : \nabla\nabla r_i)\sigma_i^2 + 2\sigma_i^2 \nabla\nabla r_i \sigma_i^2)$$

$$- \frac{n_i^2}{2\sqrt{(2\pi)^d} \left| \sqrt{2}\sigma_i \right|} \left( \sigma_i^2 b_i + \frac{1}{4}(\sigma_i^2 : \nabla\nabla b_i)\sigma_i^2 + \frac{1}{2} \sigma_i^2 \nabla\nabla b_i \sigma_i^2 \right)$$

$$+ n_i \sigma_i^2 \sum_j \left( \alpha_{i,j} + \frac{1}{2} \nabla_j \nabla_j \alpha_{i,j} : \sigma_j^2 \right) n_j$$

$$+ \frac{n_i}{2} \sum_j ((\sigma_i^2 : (\nabla_i \nabla_i \alpha_{i,j}))\sigma_i^2 + 2\sigma_i^2 (\nabla_i \nabla_i \alpha_{i,j})\sigma_i^2)n_j + 2n_i g_i I.$$

By subtracting the equation for $\sigma_i^2(dn_i/dt)$ from A.4.1, we then obtain

$$n_i \frac{d}{dt} \sigma_i^2 = n_i(\sigma_i^2 \nabla\nabla r_i \sigma_i^2) - \frac{n_i^2}{2\sqrt{(2\pi)^d} \left| \sqrt{2}\sigma_i \right|} \left( -\sigma_i^2 b_i - \frac{1}{4}(\sigma_i^2 : \nabla\nabla b_i)\sigma_i^2 + \frac{1}{2} \sigma_i^2 \nabla\nabla b_i \sigma_i^2 \right)$$

$$+ n_i \sigma_i^2 \left( \sum_j (n_j \nabla_i \nabla_i \alpha_{i,j}) \right) \sigma_i^2 + 2n_i g_i I,$$

which can be grouped as

$$\frac{d}{dt} \sigma_i^2 = V_0 + V_{1,i} \sigma_i^2 + \sigma_i^2 V_{2,i} \sigma_i^2.$$

Where the three terms represent a driver for increasing trait variance due to intergenerational genetic mutations

$$V_{0,i} = 2g_i I,$$

a damping term associated with individual self-limitation

$$V_{1,i} = \frac{n_i}{2\sqrt{(2\pi)^d}\left|\sqrt{2}\sigma_i\right|}\left(b_i + \frac{1}{2}(\sigma_i^2 : \nabla\nabla b_i)\right)$$

and a damping term related to the curvature of the fitness landscape and intraspecific competition

$$V_{2,i} = \nabla\nabla r_i + \sum_j \nabla_i\nabla_i\alpha_{i,j}n_j - \frac{n_i\nabla\nabla b_i}{2\sqrt{(2\pi)^d}\left|\sqrt{2}\sigma_i\right|}.$$

# Appendix B. Definition of parameters for numerical example

Constant parameters

$$b(x,t) = 10^{-3}, \quad \alpha(x,x') = -1, \quad g(x) = 2\cdot10^{-5}.$$

Temporal change of net *per capita* growth rate:

$$r(x,t) = 1 - (1 - f(t))r_0(x) + f(t)r_\infty(x).$$

Here, the function $f(t)$ provides a gradual transition from the initial state to the final state, given by

$$f(t) = \begin{cases} t/2000 & \text{if} \quad t \le 2000 \\ 1 & \text{if} \quad t > 2000 \end{cases}.$$

The initial and final states of the net growth rates

$$r_0(x) = \left\|x - \left[\frac{1}{2},\frac{1}{2}\right]\right\|^2, \quad r_\infty(x) = 4\left\|x - \left[\frac{3}{4},\frac{1}{4}\right]\right\|^2\left\|x - \left[\frac{1}{4},\frac{3}{4}\right]\right\|^2.$$

Initial conditions for the single-species simulation

$$n_1(0) = 1, \quad x_1(0) = [0.6, 0.4] \quad \text{and} \quad \sigma_1^2(t) = 0.005\,I.$$

Initial conditions for the two-species simulation

$$x_1(0) = [0.6, 0.4] \quad \text{and} \quad x_2(0) = [0.4, 0.6]$$

and

$$n_1(0) = n_2(0) = 0 \quad \text{and} \quad \sigma_1^2(0) = \sigma_2^2(0) = 0.005\,I.$$

For completeness, we also give the Hessian matrix of $r(x,t)$ at $[1/2, 1/2]$

$$\nabla\nabla r|_{[1/2,\ 1/2]} = -2(1 - f(t))\begin{bmatrix} 1 & 0 \\ 0 & 1 \end{bmatrix} + 2f(t)\begin{bmatrix} 0 & 1 \\ 1 & 0 \end{bmatrix},$$

which has eigenvalues $-2$ and $-2 + 4f(t)$. In particular, the maximum becomes a saddle point when $f(t) = 1/2$, i.e. at $t = 1000$.

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
