## [Reviewer comments · Royal Society Open Science]

Review History

RSOS-200321.R0 (Original submission)

Review form: Reviewer 1

Is the manuscript scientifically sound in its present form?

Yes

Are the interpretations and conclusions justified by the results?

Yes

Is the language acceptable?

Yes

Do you have any ethical concerns with this paper?

No

Have you any concerns about statistical analyses in this paper?

No

Recommendation?

Major revision is needed (please make suggestions in comments)

Comments to the Author(s)

In this manuscript a PDE model describing densities of individuals in trait space (“individual-based model”) is approximated by an ODE model for the abundance of individuals, mean trait and trait variance (“species-level model”). The model assumptions include a net growth term, intra- and inter-specific competition and inheritance (trait diffusion).

This is definitely an interesting approach and I appreciate the effort of relating deterministic individual and species-based models. However, the manuscript would benefit from some clarifications. There are three major issues that I believe should be addressed: 1. putting it more into context with the existing literature, 2. clarifying definitions and relating the formulas more to biology and vice versa and 3. discussing the assumption of the normal distribution over traits in more detail.

Major suggestions:

1. context and citations: The manuscript is put in the context of adaptive dynamics. Since it is in particular about individual vs species descriptions it should be put into a broader context. Please relate your work to major approaches and results from evolutionary game theory, population genetics and other eco-evolutionary modelling. Compare it to other individual-based models and agent based models (to me an individual-based model requires a discrete number of individuals that interact). Also the relation to birth-death processes with finite/infinite numbers of individuals would help the readers understand the context. I suggest the following literature:

- Grimm et al. (ABM, IBM) eg. DeAngelis, Grimm in F1000 reports 2014
- Black and McKane TREE 2012 (IBM)
- (some examples of IBM → species and eco-evo: Huang et al. PNAS 2015, Schenk et al. BMC Evol Bio 2020)
- Claussen 2008: STOCHASTIC MODELS IN BIOLOGICAL SCIENCES BANACH CENTER PUBLICATIONS, VOLUME 80
- Czuppon, Traulsen arXiv 2019 (finite, infinite, see also refs therein)
- Sharkey, J mathem. biology, 2008 (deterministic individual vs species-level models)
- (and in comparison to ref [6] it would be very insightful to compare it to Veller et al. PNAS 2017)

2. Accessibility, words and definitions: please make the paper accessible to a wider audience. The usage of key terms from biology like monomorphic/polymorphic vs individual/species or fitness need more explanation.

- In general the manuscript would benefit from a more structured introduction into the 3 models. Maybe a schematic visualisation that includes all three “models” shown in Fig 4. (e.g. PDE – Taylor, moment closure.. → ODE and PDE – numerical integration → trait mean, var model). This would also include choosing one name for a model. At the moment switching between different terms (species-level, ODE etc.) confuses.
- the authors use words from biology (individual/species/mono-/polymorphic) to categorise their models, but it is not clear how this translates to the mathematical assumptions.
- on the other hand, it is unclear what some mathematical assumptions imply for the biology.
- Dieckmann et al. do not consider species-level models as strictly polymorphic (page 3, line 16)
- why is the deterministic-individual category considered individual-based? The model PDE clearly describes a whole population not an individual. Individual-based, in my understanding, is a model where interactions between discrete individuals are the basis of the process. Example: individual-based or agent-based models or cellular automata
- please explain your definition of fitness. compare with Doebeli et al. eLife 2017

3. The normal distribution assumption: The assumption of a normal distribution of traits is used throughout the manuscript. The authors should clearly state why such an assumption is meaningful biologically. Also, starting with a normal distribution and then noting that the approximation of a normal distribution corresponds well is circular. It should be shown how the models correspond when the initial distribution is other than normal, for example by starting with a random initial condition in the PDE model.

Minor suggestions: (page refers to page .. out of 29)

- page 2 line 35: "the" literature
- page 2 line 50: many "of" - difficult to read
- page 3, line 12: "a" review - not clear that [1] is meant
- page 3, line 56: for an introduction it would be helpful to cite newer literature that includes the advancement in the field and points at some examples in eco-evo models over the last 30 years.
- page 4, line 3: Dieckmann et al. do not show a direct link between monomorphic and polymorphic deterministic models, how does this translate into the link between individual and species level models?
- page 5 line 33: it is not so clear what further assumptions about the population are made "asexual" is one, yet several more come to mind such as non-spatial, not stochastic, haploid
- page 6 line 10. the role of g , particularly later in the paper, is not clear
- page 6 line 38: a single species is not defined as S groups but there are S number of species, please explain more clearly
- page 8, line 44: why must there be a trade-off? can a species have high growth rate and high variance? please explain.
- page 8, line 55-60: in A4.1. you name the terms self- and intra- here it is intra- and inter-. But there is no inter-specific term
- page 9, line 3: please make the statement, it's not too obvious
- page 9 line 19: interpretation of eq 2.9 is missing
- page 10 line 33: "second (ii)" ?
- page 10 line 56: what is the individual fitness function
- page 11, line 49: what is the fitness landscape
- page 13 line 9: is this the euclidean norm? please define
- page 14 line 21: close bracket
- page 14 line 8-28: this could already be mentioned in the introduction for the derivation (sections 2.1. and 2.2), which would put the derivation into context much earlier.
- page 15 line 14: are the ecological dynamics in equilibrium at $t=0$? The initial configuration seems to be the choice of the authors (see p13 112). Since the initial choice is a normal distribution it is no surprise that an approximation using a normal distribution corresponds to that... What happens if the initial distribution is not normal. This would be interesting (see also major points above)
- page 15 line 44: what if this scaling can not be done under the assumption of the normal distribution?
- page 19, line 50: what real system could one consider for this model in general?
- page 20 line 12: "in the literature", also: please compare to other literature, see major point 1
- page 20, line 42: please explain this balance (or explain what eq (2.14) means biologically in more detail)
- page 23, line 30: what does this mean biologically (in words)
- page 23 line 41: please refer to a discussion on the normal distribution assumption
- page 26 line 14: first step is not clear, to me at least
- page 26 line 28: $\mu_{\{3, i\}}$ is not defined
- page 20, line 43: how is $r(t)$ defined mathematically for $t > 0$
- ref [6] incorrect title!?
- please provide the code for your simulations

Figure 1:

- abbreviation ODE not explained in caption
- stochastic forcing should be explained

Figure 2:

- a 2D figure would be sufficient and in my opinion easier to follow. but I recognise that this is subjective

Figure 3:

- not referenced in text

Figure 4:

- what does the colour represent?
- the figures are very dark
- please refer to equations when talking about the "PDE" or "ODE" model.
- what equations/model generate the middle plane? is that 2.2., 2.3., 2.4?
- the caption should contain all of the information. what time point is this, what are the parameter values, what program was used
- most clear if labels on plot panes
- especially here, a 3D projection does not help much. 3 subplots with colouring that show hight would be more informative, especially because colour has no meaning here

Figure 5:

- consistent colouring would help the reader x and σ have different colours compared to Fig 3

Figure 6:

- Colour inconsistent. It would be better to use a colour-map than colours used for something else (Fig 3, 5) previously.
- plot/visualise $r(t)$ either in the plot or in a subplot. It is described in the text but maybe lost to the reader that the dynamics are extremely forced by $r(t)$.

Figure 7:

- labels as in text missing: eg. 'single species model' and 'PDE model'
- can you also plot (as written in line 18) eq 2.11?
- this figure shows x_1 , with $x_1(0)=0.6$ not x_2 , with $x_2(0)=0.4$, correct? This should be in the caption. Same for all the other figures and measures

Figure 8:

- Do the variances σ_1 and σ_2 exactly overlap? How is the variance plotted if it is a 2x2 matrix?...
- Can a Figure similar to Fig 6 be beneficial for this case?

Figure 9:

- see Fig 4 comments
- here the plot would really benefit from a 2D contour representation to see that the normal distribution assumption is not fulfilled
- what time point is shown?

Review form: Reviewer 2 (Axel Rossberg)

Is the manuscript scientifically sound in its present form?

Yes

Are the interpretations and conclusions justified by the results?

Yes

Is the language acceptable?

Yes

Do you have any ethical concerns with this paper?

No

Have you any concerns about statistical analyses in this paper?

No

Recommendation?

Reject

Comments to the Author(s)

In this manuscript the dynamics of mean and variance of the traits of evolving species are derived from a deterministic approximation of the dynamics of the density of self-reproducing individuals in trait space.

The paper's idea and approach has much in common with a paper by Sasaki and Dieckmann from 2010 (<http://dx.doi.org/10.1007/s00285-010-0380-6>), which, unfortunately, the authors appear to have missed. It can't be the reviewer's role to tease out the fine points by which the two theories might differ, and I shall not try this here.

However, it is worth noting that both theories share the same flaw: both appear to ignore that in reality finite population size and the resulting demographic stochasticity play a decisive roles in determining the distribution of individuals in trait space (<http://dx.doi.org/10.1098/rspb.2013.1248>), genetic variance (<http://doi.org/10.1209/0295-5075/97/40008>), and even selection gradients (<http://dx.doi.org/10.1073/pnas.1603693113>) --- at least in the case of clonal reproduction considered here.

A revised manuscript would need to convincingly address both points.

Review form: Reviewer 3

Is the manuscript scientifically sound in its present form?

Yes

Are the interpretations and conclusions justified by the results?

Yes

Is the language acceptable?

Yes

Do you have any ethical concerns with this paper?

No

Have you any concerns about statistical analyses in this paper?

No

Recommendation?

Major revision is needed (please make suggestions in comments)

Comments to the Author(s)

This paper derives moment dynamics for a general eco-evolutionary model defined on multi-dimensional trait spaces. Since the model describes the dynamics of phenotype distribution (driven by selection and mutation) in the form of partial differential equation, its approximation with moment dynamics in the form of ordinary differential equations enhances analytical and numerical tractability. I have three comments below.

[Major comments]

1. Debrre et al. (2014) have described the dynamics of phenotype distribution (through selection) in multi-dimensional trait spaces in a manner similar to this paper. Thus, clarifying the difference of this paper from Debreer et al. (2014) is needed.

Debarre F., S. L. Nuismer & M. Doebeli. 2014. Multidimensional (co)evolutionary stability. *Am. Nat.* 184: 158-171.

2. In "Numerical examples", authors seem to assume a static saddle shape for the fitness landscape (while its height can change). Thus, the saddle point is not convergence stable. In this case, for occurrence of diversifying evolution, called speciation in this paper, the initial population may be required to be sufficiently close to the saddle point as well as has a sufficiently large variance. Otherwise, the initial population just directionally evolves to either of the two upper parts of the saddle. Therefore, if authors could present another example in which the trait space has a point that is not only evolutionary unstable (described in this paper as the exponential growth of variance) but also convergence stable (e.g. Vukics et al. 2003, see also Geritz et al. 2016), this section would become more convincing.

Vukics A., J. Asboth & G. Meszéna. 2003. Speciation in multidimensional evolutionary space. *Phys. Rev. E* 68:41903

Geritz S. A. H., J. A. J. Metz & C. Rueffler. 2016. Mutual invadability near evolutionarily singular strategies for multivariate traits, with special reference to the strongly convergence stable case. *J. Math. Biol.* 72: 1081-1099.

3. Figure 8 shows comparison of evolutionary trajectories of species in individual-level model (partial differential equation) with those in species-level model (moment dynamics approximation). I think this is an important result. The amount of discrepancy would depend on the timing of replacing a single species with two slightly different species, as well as their initial slight difference. To my knowledge, how to choose appropriate timing and initial difference is still an open question. If authors could propose (or discuss) the optimal timing and initial difference such that they can minimize the discrepancy, it would increase the importance of this paper.

[Minor comments]

Eq. (2.6): Please describe the definition of the binary operator ":".

Page 21 Line 16-21:

The two papers [14] and [15] in References are not cited in the main text?

Decision letter (RSOS-200321.R0)

Dear Dr Nordbotten,

The editors assigned to your paper ("The dynamics of trait variance in multi-species communities") have now received comments from reviewers. We would like you to revise your paper in accordance with the referee and Associate Editor suggestions which can be found below (not including confidential reports to the Editor). Please note this decision does not guarantee eventual acceptance.

Please submit a copy of your revised paper before 02-Jul-2020. Please note that the revision deadline will expire at 00.00am on this date. If we do not hear from you within this time then it will be assumed that the paper has been withdrawn. In exceptional circumstances, extensions

may be possible if agreed with the Editorial Office in advance. We do not allow multiple rounds of revision so we urge you to make every effort to fully address all of the comments at this stage. If deemed necessary by the Editors, your manuscript will be sent back to one or more of the original reviewers for assessment. If the original reviewers are not available, we may invite new reviewers.

- Data accessibility

If you wish to submit your supporting data or code to Dryad (<http://datadryad.org/>), or modify your current submission to dryad, please use the following link:
<http://datadryad.org/submit?journalID=RSOS&manu=RSOS-200321>

- Competing interests

- Authors' contributions

- Acknowledgements

- Funding statement

on behalf of Professor Tim Rogers (Associate Editor) and Mark Chaplain (Subject Editor)
openscience@royalsociety.org

Reviewers' Comments to Author:

Reviewer: 1

Comments to the Author(s)

In this manuscript a PDE model describing densities of individuals in trait space ("individual-based model") is approximated by an ODE model for the abundance of individuals, mean trait and trait variance ("species-level model"). The model assumptions include a net growth term, intra- and inter-specific competition and inheritance (trait diffusion).

This is definitely an interesting approach and I appreciate the effort of relating deterministic individual and species-based models. However, the manuscript would benefit from some clarifications. There are three major issues that I believe should be addressed: 1. putting it more into context with the existing literature, 2. clarifying definitions and relating the formulas more to biology and vice versa and 3. discussing the assumption of the normal distribution over traits in more detail.

Major suggestions:

1. context and citations: The manuscript is put in the context of adaptive dynamics. Since it is in particular about individual vs species descriptions it should be put into a broader context. Please relate your work to major approaches and results from evolutionary game theory, population genetics and other eco-evolutionary modelling. Compare it to other individual-based models and agent based models (to me an individual-based model requires a discrete number of individuals that interact). Also the relation to birth-death processes with finite/infinite numbers of individuals would help the readers understand the context. I suggest the following literature:

- Grimm et al. (ABM, IBM) eg. DeAngelis, Grimm in F1000 reports 2014

- Black and McKane TREE 2012 (IBM)

- (some examples of IBM → species and eco-evo: Huang et al. PNAS 2015, Schenk et al. BMC Evol Bio 2020)

- Claussen 2008: STOCHASTIC MODELS IN BIOLOGICAL SCIENCES BANACH CENTER PUBLICATIONS, VOLUME 80
- Czuppon, Traulsen arXiv 2019 (finite, infinite, see also refs therein)
- Sharkey, J mathem. biology, 2008 (deterministic individual vs species-level models)
- (and in comparison to ref [6] it would be very insightful to compare it to Veller et al. PNAS 2017)

2. Accessibility, words and definitions: please make the paper accessible to a wider audience. The usage of key terms from biology like monomorphic/polymorphic vs individual/species or fitness need more explanation.

- In general the manuscript would benefit from a more structured introduction into the 3 models. Maybe a schematic visualisation that includes all three “models” shown in Fig 4. (e.g. PDE – Taylor, moment closure.. → ODE and PDE – numerical integration → trait mean, var model). This would also include choosing one name for a model. At the moment switching between different terms (species-level, ODE etc.) confuses.
- the authors use words from biology (individual/species/mono-/polymorphic) to categorise their models, but it is not clear how this translates to the mathematical assumptions.
- on the other hand, it is unclear what some mathematical assumptions imply for the biology.
- Dieckmann et al. do not consider species-level models as strictly polymorphic (page 3, line 16)
- why is the deterministic-individual category considered individual-based? The model PDE clearly describes a whole population not an individual. Individual-based, in my understanding, is a model where interactions between discrete individuals are the basis of the process. Example: individual-based or agent-based models or cellular automata
- please explain your definition of fitness. compare with Doebeli et al. eLife 2017

3. The normal distribution assumption: The assumption of a normal distribution of traits is used throughout the manuscript. The authors should clearly state why such an assumption is meaningful biologically. Also, starting with a normal distribution and then noting that the approximation of a normal distribution corresponds well is circular. It should be shown how the models correspond when the initial distribution is other than normal, for example by starting with a random initial condition in the PDE model.

Minor suggestions: (page refers to page .. out of 29)

- page 2 line 35: “the” literature
- page 2 line 50: many “of” - difficult to read
- page 3, line 12: “a” review - not clear that [1] is meant
- page 3, line 56: for an introduction it would be helpful to cite newer literature that includes the advancement in the field and points at some examples in eco-evo models over the last 30 years.
- page 4, line 3: Dieckmann et al. do not show a direct link between monomorphic and polymorphic deterministic models, how does this translate into the link between individual and species level models?
- page 5 line 33: it is not so clear what further assumptions about the population are made “asexual” is one, yet several more come to mind such as non-spatial, not stochastic, haploid
- page 6 line 10. the role of g , particularly later in the paper, is not clear
- page 6 line 38: a single species is not defined as S groups but there are S number of species, please explain more clearly
- page 8, line 44: why must there be a trade-off? can a species have high growth rate and high variance? please explain.
- page 8, line 55-60: in A4.1. you name the terms self- and intra- here it is intra- and inter-. But there is no inter-specific term
- page 9, line 3: please make the statement, it’s not too obvious
- page 9 line 19: interpretation of eq 2.9 is missing
- page 10 line 33: “second (ii)” ?
- page 10 line 56: what is the individual fitness function
- page 11, line 49: what is the fitness landscape
- page 13 line 9: is this the euclidean norm? please define

- page 14 line 21: close bracket
- page 14 line 8-28: this could already be mentioned in the introduction for the derivation (sections 2.1. and 2.2), which would put the derivation into context much earlier.
- page 15 line 14: are the ecological dynamics in equilibrium at $t=0$? The initial configuration seems to be the choice of the authors (see p13 112). Since the initial choice is a normal distribution it is no surprise that an approximation using a normal distribution corresponds to that... What happens if the initial distribution is not normal. This would be interesting (see also major points above)
- page 15 line 44: what if this scaling can not be done under the assumption of the normal distribution?
- page 19, line 50: what real system could one consider for this model in general?
- page 20 line 12: "in the literature", also: please compare to other literature, see major point 1
- page 20, line 42: please explain this balance (or explain what eq (2.14) means biologically in more detail)
- page 23, line 30: what does this mean biologically (in words)
- page 23 line 41: please refer to a discussion on the normal distribution assumption
- page 26 line 14: first step is not clear, to me at least
- page 26 line 28: $\mu_{\{3, i\}}$ is not defined
- page 20, line 43: how is $r(t)$ defined mathematically for $t>0$
- ref [6] incorrect title?
- please provide the code for your simulations

Figure 1:

- abbreviation ODE not explained in caption
- stochastic forcing should be explained

Figure 2:

- a 2D figure would be sufficient and in my opinion easier to follow. but I recognise that this is subjective

Figure 3:

- not referenced in text

Figure 4:

- what does the colour represent?
- the figures are very dark
- please refer to equations when talking about the "PDE" or "ODE" model.
- what equations/model generate the middle plane? is that 2.2., 2.3., 2.4?
- the caption should contain all of the information. what time point is this, what are the parameter values, what program was used
- most clear if labels on plot panes
- especially here, a 3D projection does not help much. 3 subplots with colouring that show height would be more informative, especially because colour has no meaning here

Figure 5:

- consistent colouring would help the reader x and σ have different colours compared to Fig 3

Figure 6:

- Colour inconsistent. It would be better to use a colour-map than colours used for something else (Fig 3, 5) previously.
- plot/visualise $r(t)$ either in the plot or in a subplot. It is described in the text but maybe lost to the reader that the dynamics are extremely forced by $r(t)$.

Figure 7:

- labels as in text missing: eg. 'single species model' and 'PDE model'
- can you also plot (as written in line 18) eq 2.11?
- this figure shows x_1 , with $x_1(0)=0.6$ not x_2 , with $x_2(0)=0.4$, correct? This should be in the caption. Same for all the other figures and measures

Figure 8:

- Do the variances σ_1 and σ_2 exactly overlap? How is the variance plotted if it is a 2x2 matrix?...

- Can a Figure similar to Fig 6 be beneficial for this case?

Figure 9:

- see Fig 4 comments

- here the plot would really benefit from a 2D contour representation to see that the normal distribution assumption is not fulfilled

- what time point is shown?

Reviewer: 2

Comments to the Author(s)

In this manuscript the dynamics of mean and variance of the traits of evolving species are derived from a deterministic approximation of the dynamics of the density of self-reproducing individuals in trait space.

The paper's idea and approach has much in common with a paper by Sasaki and Dieckmann from 2010 (<http://dx.doi.org/10.1007/s00285-010-0380-6>), which, unfortunately, the authors appear to have missed. It can't be the reviewer's role to tease out the fine points by which the two theories might differ, and I shall not try this here.

However, it is worth noting that both theories share the same flaw: both appear to ignore that in reality finite population size and the resulting demographic stochasticity play a decisive roles in determining the distribution of individuals in trait space (<http://dx.doi.org/10.1098/rspb.2013.1248>), genetic variance (<http://doi.org/10.1209/0295-5075/97/40008>), and even selection gradients (<http://dx.doi.org/10.1073/pnas.1603693113>) --- at least in the case of clonal reproduction considered here.

A revised manuscript would need to convincingly address both points.

Reviewer: 3

Comments to the Author(s)

This paper derives moment dynamics for a general eco-evolutionary model defined on multi-dimensional trait spaces. Since the model describes the dynamics of phenotype distribution (driven by selection and mutation) in the form of partial differential equation, its approximation with moment dynamics in the form of ordinary differential equations enhances analytical and numerical tractability. I have three comments below.

[Major comments]

1. Debrre et al. (2014) have described the dynamics of phenotype distribution (through selection) in multi-dimensional trait spaces in a manner similar to this paper. Thus, clarifying the difference of this paper from Debrre et al. (2014) is needed.

Debarre F., S. L. Nuismer & M. Doebeli. 2014. Multidimensional (co)evolutionary stability. *Am. Nat.* 184: 158-171.

2. In "Numerical examples", authors seem to assume a static saddle shape for the fitness landscape (while its height can change). Thus, the saddle point is not convergence stable. In this case, for occurrence of diversifying evolution, called speciation in this paper, the initial population may be required to be sufficiently close to the saddle point as well as has a sufficiently large variance. Otherwise, the initial population just directionally evolves to either of the two upper parts of the saddle. Therefore, if authors could present another example in which the trait space has a point that is not only evolutionary unstable (described in this paper as the exponential growth of variance) but also convergence stable (e.g. Vukics et al. 2003, see also Geritz et al. 2016), this section would become more convincing.

Vukics A., J. Asboth & G. Meszéna. 2003. Speciation in multidimensional evolutionary space. *Phys. Rev. E* 68:41903

Geritz S. A. H., J. A. J. Metz & C. Rueffler. 2016. Mutual invadability near evolutionarily singular strategies for multivariate traits, with special reference to the strongly convergence stable case. *J. Math. Biol.* 72: 1081-1099.

3. Figure 8 shows comparison of evolutionary trajectories of species in individual-level model (partial differential equation) with those in species-level model (moment dynamics approximation). I think this is an important result. The amount of discrepancy would depend on the timing of replacing a single species with two slightly different species, as well as their initial slight difference. To my knowledge, how to choose appropriate timing and initial difference is still an open question. If authors could propose (or discuss) the optimal timing and initial difference such that they can minimize the discrepancy, it would increase the importance of this paper.

[Minor comments]

Eq. (2.6): Please describe the definition of the binary operator "·".

Page 21 Line 16-21:

The two papers [14] and [15] in References are not cited in the main text?

Author's Response to Decision Letter for (RSOS-200321.R0)

See Appendix A.

Decision letter (RSOS-200321.R1)

Dear Dr Nordbotten,

It is a pleasure to accept your manuscript entitled "The dynamics of trait variance in multi-species communities" in its current form for publication in Royal Society Open Science.

Please ensure that you send to the editorial office an editable version of your accepted manuscript, and individual files for each figure and table included in your manuscript. You can send these in a zip folder if more convenient. Failure to provide these files may delay the processing of your proof.

on behalf of Professor Tim Rogers (Associate Editor) and Mark Chaplain (Subject Editor)
openscience@royalsociety.org

Appendix A

Jan Martin Nordbotten
Department of Mathematics
University of Bergen
PB 7200
5020 BERGEN, Norway
jan.nordbotten@math.uib.no

Dear Editors of RSOS,

Thank you for considering our manuscript “The dynamics of trait variance in multi-species communities”, and for soliciting three careful reviews. We are very much aware of the challenges in securing good reviews in these times.

Our overall summary of the reviews and our revision is as follows: The reviewers do not criticize our scientific results and derivations, hence we have not made substantial changes to the mathematical derivations. On the other hand, the reviewers all ask for a clarification of our paper relative to existing literature. In particular, both reviewer #2 and #3 point to papers that are similar in aims and scope to ours. We have carefully considered all the suggested references by the reviewers, and included the majority of them in our revision. Moreover, the reviewers had several suggestions how to improve and clarify the presentation. We have incorporated these suggestions in the revision.

A detailed response to each of the reviewers’ comments is included on the following pages, and a version of the manuscript with changes highlighted is included with the submission of this revised manuscript.

In order to comply with the policy of the journal that all computer code be made publically available, we have prepared a clean version of the computer code used in the examples, which has been uploaded as electronic supplementary material with this submission.

We are confident that the revised manuscript is a significant improvement over the original submission, and hope you will find it suitable for RSOS.

Best regards on behalf of the authors,

Jan M. Nordbotten

Response to Reviewer #1

We thank the reviewer for considering our manuscript, and for pointing out interesting references and potential for improvements. We also thank the reviewer for meticulously listing minor comments, which have been implemented in the revision as best we could. We have addressed the major comments of the reviewer comments as follows:

Major comment #1: *context and citations: The manuscript is put in the context of adaptive dynamics. Since it is in particular about individual vs species descriptions it should be put into a broader context. (...)*

Response: We thank the reviewer for literature suggestions, including some very recent work.

Changes made: We have included several of the references in the introduction, which has been expanded in several places. Moreover, (and as also indicated in the reply to comment #2), we have avoided the descriptor “individual-based” model for the fine-scale model, in order to avoid any confusion with explicit models of individual behavior such as cellular automata etc. As a consequence, we have also chosen not to discuss these models to any extent in the introduction.

Major comment #2: *Accessibility, words and definitions: please make the paper accessible to a wider audience. The usage of key terms from biology like monomorphic/polymorphic vs individual/species or fitness need more explanation. (...)*

Response: We thank the reviewer for the suggestions. In the revision, we have tried to clarify the terms and their implications as much as possible.

Changes made: We have attempted to further clarify our terminology, and used a consistent terminology throughout the manuscript. Responding in particular to the comments of this reviewer, we have:

- a. Consistently referred to the fine-scale model as “population-level model” and the upscaled model as “species-level model” with abbreviations PLM and SLM used in section 4.
- b. Clarified that we do not use a precise fitness concept, since one of our observations is that the classical notion that the rate of trait evolution is proportional to the product of variance and fitness gradient is not in general accurate.
- c. The three main assumptions are summarized in biological terms in the first paragraph of section 3.
- d. We have reworded the motivation for altering the Dieckmann figure in a way that does not imply that they consider only monomorphic species models.

Major comment #3: *The normal distribution assumption: The assumption of a normal distribution of traits is used throughout the manuscript. The authors should clearly state why such an assumption is meaningful biologically. Also, starting with a normal distribution and then noting that the approximation of a normal distribution corresponds well is circular. It should be shown how the models correspond*

when the initial distribution is other than normal, for example by starting with a random initial condition in the PDE model.

Response: The reviewer here raises two separate points. One is the use of the normal distribution as a closure approximation in the derivations. This is a well-established methodology, and is justified (both in biology and otherwise) as suitable whenever a law of large numbers or diffusive process can be considered as an applicable approximation, which then leads to the normal distribution. The second point raised by the reviewer, regarding a circular argument, we infer refers to the numerical example (certainly we do not start with a normal distribution in the derivations). In this case, we use an identical initial condition between the two models in order to make a fair comparison of whether the upscaled equations are in agreement with the original equations. Unlike the impression that we unfortunately communicated to the reviewer, we do not conclude that the normal distribution is a good approximation. Indeed, we argue that skewness of the distribution is probably the main reason why there is a disparity in the velocity of traits between the models, and it is clearly visible in Figures 4, 6 and 9 that the true distribution deviates substantially from a normal distribution.

Changes made: We have clarified the use and implication of the normal distribution throughout the manuscript. Additionally, we have added a discussion of other closure models after equation (2.9), and also added a new section A.2.2 to complement this discussion. Finally, when discussing the numerics, we have emphasized that while the initial condition is a normal distribution, the solution $n(x, t)$ of the population-level model quickly deviates somewhat from a normal distribution.

Notes to minor comments:

- We have added clarifications to all questions raised by the reviewer.
- We have uploaded the computer code with the revision.
- We have critically reviewed all figures in the paper, in particular with respect to the use of colors. We hope the reviewer will find the revised figures an improvement.
- In revising the figures, we have paid particular attention to improving figures 4 and 9 as suggested by the reviewer. Our opinion is that these figures better complement the 2D projection (Figure 6) and time-evolutions (Figures 5, 7, and 8) than a contour plot would.
- We have, however, not followed the suggestion that all relevant information for the figure be presented in the caption (“*what time point is this, what are the parameter values, what program was used*”), as our opinion is that the parameters and computer program are too extensive for a caption, and are well described in the main text.
- We have also not followed the suggestion of adding a plot of $r(t)$, as we believe it is adequately covered by the description in the text, Appendix B, and the marks in Figure 6.

Response to Reviewer #2

We thank the reviewer for considering our manuscript, and for pointing out interesting references and potential for improvements. We have addressed the reviewer comments as follows:

Major Comment #1: *The paper's idea and approach has much in common with a paper by Sasaki and Dieckmann from 2010 (<http://dx.doi.org/10.1007/s00285-010-0380-6>), which, unfortunately, the authors appear to have missed. It can't be the reviewer's role to tease out the fine points by which the two theories might differ, and I shall not try this here.*

Response: There are certainly similarities between the papers, when considering them superficially, and we thank the reviewer for pointing out this manuscript to us, which we were not aware of. That said, the actual context and the methods used in the paper are quite different, not just in the “fine points”.

Firstly, the S&D paper considers a much simpler setting than ours: a single trait and a simpler evolution equation on the fine scale (compare their equation (1) and our (2.1)). Secondly, the S&D paper considers different morphs within the same species, while we consider different species. This is illustrated in the diagram below.

This allows for different technical tools, based on different assumptions, and leads to different results: S&D rely primarily on Taylor expansions, while we use integrals. As a consequence, the results of the S&D paper hold up to e.g. $\mathcal{O}(\epsilon^3)$ (see their equation (24), where their ϵ is the distance between peaks). We have not provided formal truncation errors in our manuscript (these are easily gleaned from standard calculus), but our results have truncation errors depending on the width of each species, indicated by ϵ in the figure. Thus our results are also qualitatively different from S&D. Thirdly, the S&D derivation is based on a “definition” (this is actually an assumption), that one can define p_i such that their equation (6) holds (which it almost surely does not for solutions to their equation (1)). In contrast, our derivation is based on a stated assumption that the concept of species can be uniquely defined (see our equation (2.2)). These underlying assumptions are thus fundamentally different.

Changes made: We have included a reference to S&D in the introduction and when discussing moment closure.

Major comment #2: *However, it is worth noting that both theories share the same flaw: both appear to ignore that in reality finite population size and the resulting demographic stochasticity play a decisive roles in determining the distribution of individuals in trait space (<http://dx.doi.org/10.1098/rspb.2013.1248>), genetic variance (<http://doi.org/10.1209/0295-5075/97/40008>), and even selection gradients (<http://dx.doi.org/10.1073/pnas.1603693113>) --- at least in the case of clonal reproduction considered here.*

Response: We agree with the reviewer that demographic stochasticity may lead to outcomes that differ from the predictions of deterministic models, but it seems we disagree on how large and how common such differences are. The papers the reviewer refers to assume that population size is finite and stochastic, which is a realistic assumption. We do not assume infinite population size, but we do neglect demographic stochasticity. However, our model contains a genetic diffusion term that allows the trait values of the next generation to differ from those of the current, and may cover stochastic processes such as genetic drift that are due to finite population size. The papers the reviewer refers to show that in certain models, under certain assumptions, finite, stochastic populations may behave differently from deterministic predictions, sometimes very differently. We believe, however, that there is broad consensus that unless population size is very small and/or stochasticity particularly pronounced, there is typically little discrepancy between stochastic finite models and deterministic models.

Changes made: In the revised manuscript, we state explicitly that especially if population size is small and stochasticity pronounced, other dynamics than that predicted by our model are possible.

Response to Reviewer #3

We thank the reviewer for considering our manuscript, and for pointing out interesting references and potential for improvements. We have addressed the reviewer comments as follows:

Major comment #1: *Debrre et al. (2014) have described the dynamics of phenotype distribution (through selection) in multi-dimensional trait spaces in a manner similar to this paper. Thus, clarifying the difference of this paper from Debrre et al. (2014) is needed.*

Response: We thank the reviewer for making us aware of this very nice paper, which we unfortunately had missed. The paper certainly shares many similarities in scope, and is relevant for our study. However, there are differences between the papers that affect the results, both qualitatively and quantitatively. Most notable is the initial starting point, where the Débarre paper uses a somewhat general, but quasi-linear fine-scale model (their equation (1)), whereas we use a more specific non-linear integro-differential equation (our equation (2.1)). An aspect of the Débarre model and derivation is the requirement that fitness is “scaled to 1”, as stated before their Equation (1). This is actually quite restrictive.

This leads to several differences in the results: Firstly, for Débarre, due to their scaling, they have no population dynamics (all species have constant and unit population size). In contrast, we explicitly allow for population dynamics in our model. Secondly, no smoothness of the solution can be expected with the Débarre model, as they have no differential term in trait space (mathematically, one should expect at best L^2 regularity), in our case, a smoother solution can be expected (mathematically, we expect H^1 regularity for the weak solution, but with some assumptions on the parameters even C^∞ is possible). As a consequence, the use of Taylor expansions is strictly speaking not valid for the model Débarre considers. Thirdly, both the non-linear self-limitation term we consider (bn^2), as well as the differential term ($\nabla \cdot (g\nabla n)$) introduces complexities in the derivation, such that straight-forward “linear-like” results such as (4) and (6) of Débarre are not valid (compare to our equations (2.10) and (2.11)).

The above is illustrated for a case where our models coincide, as close as possible (i.e. the first “illustration” of Débarre with their $K(x) = 1$ compared to our model in the special case of $b = g = 0$ and $r = 1$). Now the results, fortunately, also coincide (e.g. their equation (12) compared to our equation (2.14)). However, and as noted above, the underlying model and the derivations used in the two papers allow for different generalizations.

Changes made: In the revised manuscript, we carefully address the differences between the current manuscript and Débarre et al (2014) both in the introduction, and at the end of section 2.

Major comment #2: *In "Numerical examples", authors seem to assume a static saddle shape for the fitness landscape (while its height can change). Thus, the saddle point is not convergence stable. In this case, for occurrence of diversifying evolution, called speciation in this paper, the initial population may be required to be sufficiently close to the saddle point as well as has a sufficiently large variance.*

Otherwise, the initial population just directionally evolves to either of the two upper parts of the saddle. Therefore, if authors could present another example in which the trait space has a point that is not only evolutionary unstable (described in this paper as the exponential growth of variance) but also convergence stable (e.g. Vukics et al. 2003, see also Geritz et al. 2016), this section would become more convincing.

Response: We admittedly do not fully understand this comment. An important aspect of this numerical example is that the shape of the fitness landscape is not static, but time evolving. It is precisely constructed so that the initial population is stable and reaches something resembling equilibrium early in the simulation time. Then gradually (at $t=1000$), this single equilibrium is lost, and the fitness landscape allows for two steady states. The precise time-evolution of the fitness landscape is reported in Appendix B, where it is explained how it evolves as a time-weighted sum of two different shapes. As such, we do believe that this example already as written addresses the concern of the reviewer.

Changes made: We have tried to clarify further the description of this numerical example.

Major comment #3: *Figure 8 shows comparison of evolutionary trajectories of species in individual-level model (partial differential equation) with those in species-level model (moment dynamics approximation). I think this is an important result. The amount of discrepancy would depend on the timing of replacing a single species with two slightly different species, as well as their initial slight difference. To my knowledge, how to choose appropriate timing and initial difference is still an open question. If authors could propose (or discuss) the optimal timing and initial difference such that they can minimize the discrepancy, it would increase the importance of this paper.*

Response: We fully agree with the reviewer. Indeed, in an earlier draft of this paper, we even had a section proposing exactly such a relationship as the reviewer asks for. However, proposing such a relationship without a rigorous derivation is somewhat speculative, and we realized that a proper answer to this question needs significant additional work. We are currently working on this problem, and believe that we are making good progress. However, the full analysis of speciation in these types of models requires very different, and more advanced, technical tools than those used in this paper, and therefore cannot be accommodated without lengthening the exposition beyond what is appropriate.

Changes made: We have included a discussion of speciation events, and the main challenges associated with them at the end of section 4.